# The cellular and KSHV A-to-I RNA editome in primary effusion lymphoma and its role in the viral lifecycle

Suba Rajendren[1,2,3], Xiang Ye[1,2,3], William Dunker[1,2,3], Antiana Richardson[1,2,3] & John Karijolich ®[1,2,3,4,5] ✉

Adenosine-to-inosine RNA editing is a major contributor to transcriptome diversity in animals with far-reaching biological consequences. Kaposi's sarcoma-associated herpesvirus (KSHV) is the etiological agent of several human malignancies including primary effusion lymphoma (PEL). The extent of RNA editing within the KSHV transcriptome is unclear as is its contribution to the viral lifecycle. Here, we leverage a combination of biochemical and genomic approaches to determine the RNA editing landscape in host- and KSHV transcriptomes during both latent and lytic replication in PEL. Analysis of RNA editomes reveals it is dynamic, with increased editing upon reactivation and the potential to deregulate pathways critical for latency and tumorigenesis. In addition, we identify conserved RNA editing events within a viral microRNA and discover their role in miRNA biogenesis as well as viral infection. Together, these results describe the editome of PEL cells as well as a critical role for A-to-I editing in the KSHV lifecycle.

Viruses, as obligate intracellular parasites, are exquisitely integrated into host gene expression processes and have evolved strategies to exploit many, and in some cases, all, host gene expression mechanisms. The optimization of RNA structure and function through RNA modification is an established mechanism by which organisms, and their pathogens, regulate gene expression. To date, over 170 distinct chemical modifications have been identified[1]. While significant attention has been directed towards methylated derivatives of adenosine, such as N6-methyladenosine, and viral infection[2–6], the contribution of other modifications to viral lifecycles and virus biology remains under explored.

Inosine, which results from the hydrolytic deamination of adenosine, is an abundant modification of the human transcriptome occurring at over a hundred million sites[7,8]. Adenosine-to-Inosine (A-to-I) RNA editing occurs within double-stranded RNA and is catalyzed by a family of proteins, adenosine deaminases acting on RNAs (ADARs)[8–10]. In mammals there are three highly conserved members in the ADAR family: ADAR1, ADAR2 and ADAR3[8,9]. While the enzymatic activity of ADAR1 and ADAR2 is well established, ADAR3 is catalytically-deficient, and its expression is restricted to the brain[8]. The deamination of A-to-I results in the removal of the hydrogen-donating amino group at the C6 position of adenine, leaving a hydrogen accepting oxygen, and thus changes the base-pairing potential of the nucleotide. Indeed, inosine preferentially base-pairs with cytosine. A-to-I editing can have profound effects at all stages of gene expression. Editing events within coding sequences can lead to recoding of the protein sequence, while editing sites within introns and untranslated regions (UTRs) affect pre-mRNA splicing and transcript stability, respectively[8,11]. In addition, many noncoding RNAs, including long noncoding RNA (lncRNA) and microRNAs (miRNA) are subjected to A-to-I editing with effects on small RNA-mediated gene regulation[8,12].

ADARs and A-to-I editing play a critical role in regulating host immune responses and viral replication[13]. For example, mutations within ADAR1 lead to Aicardi-Goutieres syndrome (AGS), an

[1]Department of Pathology, Microbiology, and Immunology, Vanderbilt University Medical Center, Nashville, TN 37232-2363, USA. [2]Vanderbilt Institute for Infection, Immunology and Inflammation, Nashville, TN 37232-2363, USA. [3]Vanderbilt Center for Immunobiology, Nashville, TN 37232-2363, USA. [4]Department of Biochemistry, Vanderbilt University School of Medicine, Nashville, TN 37232-2363, USA. [5]Vanderbilt-Ingram Cancer Center, Nashville, TN 37232-2363, USA. ✉e-mail: John.karijolich@vanderbilt.edu

autoimmune disease associated with spontaneous type I interferon (IFN) production[14–17]. Along this line, ADAR1 depletion leads to lethality due to MDA5 and RNAseL dependent mechanisms[18–21]. Additionally, ADAR1 knockout murine cells have reduced A-to-I editing resulting in the accumulation of immunostimulatory dsRNA that is recognized by dsRNA-activated protein kinase R (PKR), MDA5, and members of the oligoadenylate synthase (OAS) family[18,19,22,23]. A-to-I editing is thus important in the suppression of an IFN response to endogenous RNA. In contrast, the effect of ADARs on virus replication can be either antiviral or proviral and are dependent upon the specific virus and host cell combination[4,6]. Moreover, while some of the effects of ADARs on viral replication are IFN dependent, others are mediated through the effects of altered inosine base-pairing on gene expression dynamics.

Here, we describe a comprehensive atlas of A-to-I editing in the lifecycle of the oncogenic virus, Kaposi's sarcoma-associated herpesvirus (KSHV), and discover a previously unknown role for RNA editing in KSHV infection. KSHV is the etiological agent of several human malignancies, including Kaposi's sarcoma (KS) and primary effusion lymphoma (PEL)[24]. Identification of A-to-I editing events in PEL cells revealed wide-spread editing of both the host and KSHV transcriptomes preferentially occurring in intronic and untranslated regions (UTRs). KSHV lytic reactivation further expanded the host and viral editomes, revealing a dynamic A-to-I editing landscape with the potential to deregulate pathways critical for latency and tumorigenesis. Moreover, we identified three previously unknown A-to-I editing events within the KSHV miRNAs and demonstrate that editing of the KSHV miR-K12-4-3p seed region is critical for viral infection. These results reveal an important role of A-to-I editing in the KSHV lifecycle and provide a rich resource for further understanding the molecular basis of KSHV infection and KSHV-induced oncogenesis.

## Results

### Global analysis of the A-to-I editome in PEL cells

While the transcriptional landscape of latent and lytic PEL cells has been characterized by RNA-seq[25,26], determining the transcriptome during the KSHV lifecycle is challenging as upon lytic cycle induction not all cells will reactivate and thus the cell population will be a mix of both latent and lytic cells. To overcome this challenge, we modified two established PEL cell lines, TREx-BCBL1-RTA and BC3 cells, by transducing them with lentivirus harboring GFP under the control of the viral lytic gene PAN promoter (pPAN), creating TREx-BCBL1-PAN-GFP and iBC3-PAN-GFP. An additional feature of both models is the integration of a doxycycline (Dox)-inducible version of the major viral transcription activator RTA, which enables KSHV to enter its lytic cycle upon the introduction of Dox into the cell culture media. The established TREx-BCBL1-PAN-GFP and iBC3-PAN-GFP will be subsequently referred as BCBL1 and BC-3 in this manuscript. Following the induction of the lytic cycle a pure population of lytic cells can be obtained by sorting GFP+ cells (Fig. 1a). To further establish the GFP reporter system we compared the detection of lytic reactivation by two methods, RNA FLOW-FISH for the KSHV-encoded PAN RNA and GFP fluorescence (Supplementary Fig. 1a). Our data suggests that selecting GFP+ and PAN+ cells enhances selectively over sorting on only PAN RNA or GFP fluorescence (Fig S1a). 48 h post-addition of doxycycline approximately 40% of cells are both PAN+ and GFP+. This value is consistent with other studies reporting 40–60% lytic reactivation[27,28], and thus establishes the utility of our system for characterizing the KSHV transcriptome.

We collected latent (untreated) and GFP+ lytic cells 48 h post-dox induction and prepared rRNA-depleted RNA-sequencing libraries generating approximately 100 million reads per biological replicate. Principal component analysis (PCA) of latent and lytic samples revealed four distinct clusters, indicating the presence of cell-specific and infectious stage-specific transcriptome variations in PEL (Supplementary Fig. 1b). Along this line, differential gene expression analysis

using DESeq2 indicates remodeling of the PEL cell transcriptome upon lytic reactivation. (Supplementary Fig. 1c, d).

Adenosines are converted into inosines by hydrolytic deamination (Fig. 1b) and these A-to-I RNA editing sites can be identified by changes in cDNA sequence as an A-to-G mutational signature when compared to the reference genomic sequence. We performed A-to-I RNA editing site identification on the latent and lytic RNA-sequencing data using the Software for Accurately Identifying Locations Of RNA editing (SAILOR) (Fig. 1c)[29]. Briefly, A-to-G changes and T-to-C changes between the cDNA and the human reference genome were identified and sites corresponding to known human single nucleotide polymorphisms (SNPs) were excluded. The accuracy of editing events called is measured by a confidence score which is assigned using a beta distribution that considers both read depth and editing site percentage. Predicted sites with ≥99% confidence score and present in both biological replicates were used for downstream analysis.

In latent BCBL1 cells our approach identified 24,948 editing events in the host transcriptome during latent infection, whereas 36,081 editing events were identified upon lytic reactivation (Supplementary data 1) (Fig. 1d left). Similarly, 16,312 and 41,501 editing sites were identified in latent and lytic BC-3 cells, respectively (Fig. 1d right). Interestingly, the host editome of lytic cells was 1.4-fold and 2.5-fold larger than latent BCBL1 and BC-3 cells, respectively. Approximately 70% of all edited sites within both cell lines were mapped to within protein coding genes. Further analyses indicate >60% of the editing sites mapped to the protein coding transcripts fall within annotated intronic regions while ~30% of the edited sites fall within untranslated regions (UTRs). The preference for introns and UTRs is consistent with previous reports[30–34] investigating the global landscape of A-to-I editing. The intronic and UTR preference coupled with the dynamic nature of editing in lytic cells suggests A-to-I editing may co- and post-transcriptionally regulate the host response to KSHV.

Regarding editing sites that map to the non-protein coding transcriptome, greater than 50% fall within the intergenic space. To test whether the intergenic sites are just an extension of an annotated or partially annotated gene we computationally extended annotated features in the human genome 2000 base pairs up and downstream. Our data suggests that less than 20% of sites are in a 2 kb proximity of an annotated feature (Fig. 1d). Thus, the bulk of non-protein coding editing is within transcriptionally active intergenic space as well as noncoding RNAs including pseudogenes, long noncoding RNAs (lncRNAs), miRNAs and small nucleolar RNAs (snoRNAs).

A-to-I editing is a prevalent RNA modification found within transcribed Alu sequences. In fact, previous studies have reported that the majority of human A-to-I editing occurs within Alu repeats[7,35]. Through intersection analyses we investigated whether PEL cell RNA editing is similarly enriched within repetitive sequences. Consistent with previous findings, our data indicates that >90% of the A-to-I edited sites in the PEL lines tested fall within repetitive elements, with the majority located within Alu sequences (Fig. 1e). We also investigated the nucleotide context in which A-to-I editing occurs. Nearest neighbor analysis and visualization of statistically significant ($P < 0.05$) nucleotides was performed with Two Sample Logo analysis. Our data suggest that in both PEL cells the dinucleotide sequences CU and GC are overrepresented whereas AG and UA are underrepresented at the 5′ and 3′ sides of editing sites, respectively (Fig. 1f). The identified 5′ and 3′ neighboring nucleotides are consistent with previously published mammalian ADAR preferences[36].

Given the variable number of editing sites identified in BCBL1 and BC-3 cells we next sought to compare the two editomes. Intersection analyses revealed a large degree of cell-specific editing. However, these analyses did identify 5289 and 11,047 sites that were conserved in latent and lytic cells, respectively (Fig. 1g). To independently validate the editing sites, we selected three genes that displayed significant editing in their 3′ UTRs and performed Sanger sequencing on genomic

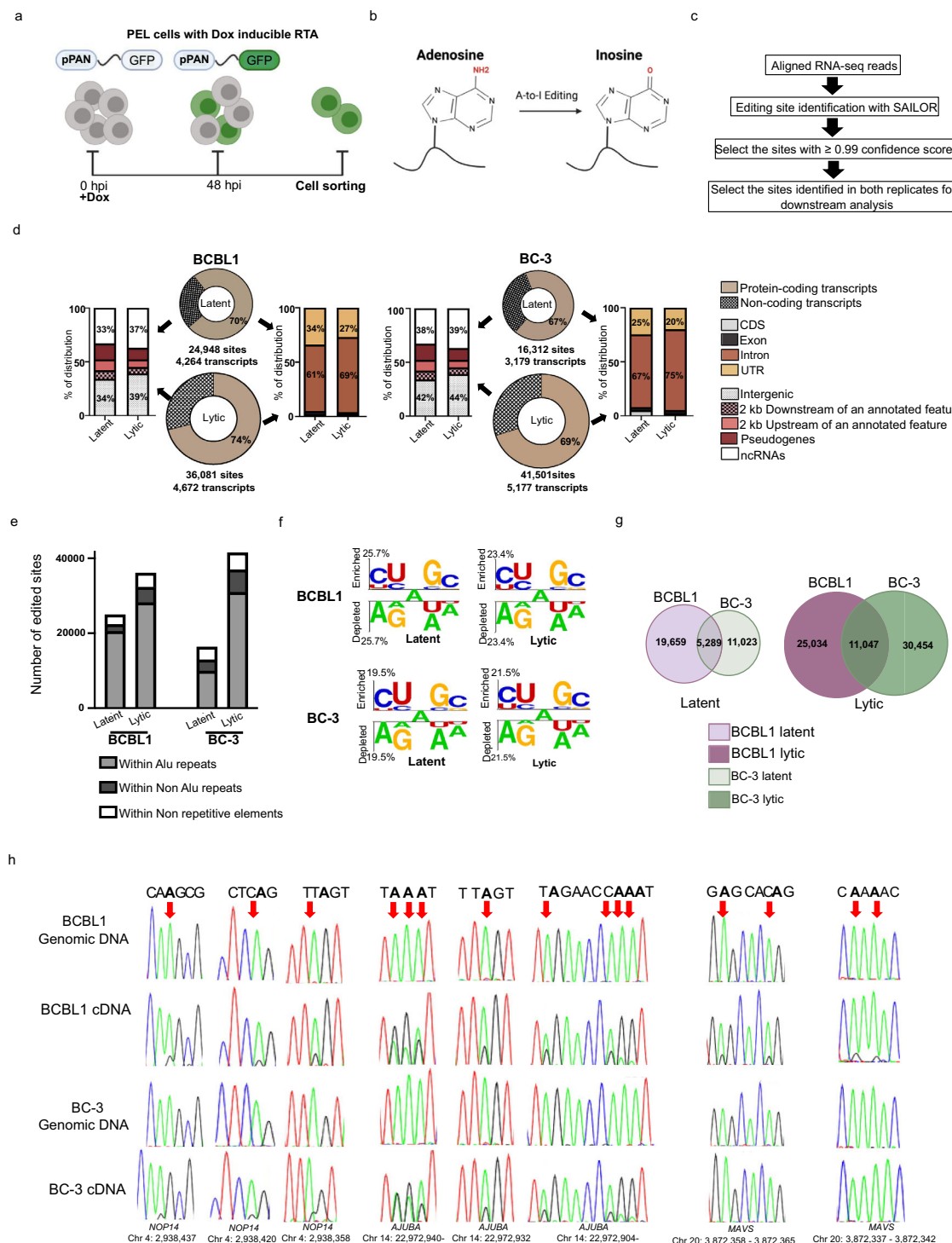

**Fig. 1 | Global analysis of host A-to-I editing during the KSHV lifecycle.**
**a** Schematic representation of the strategy to isolate lytic PEL cells by FACS. Both PEL cells have the integrated doxycycline (Dox)-inducible version of the major viral transcription activator RTA. Latent cells refer to the untreated PEL cells and lytic cells refer to the isolated GFP positive cells at 48 hpi. **b** Schematic representation of adenosine to inosine conversion by ADARs. **c** A-to-I editing analysis pipeline. **d** Genomic distribution of edited sites in BCBL1 (left) and BC-3 (right) cells. Donut charts represent the percentage of edited sites mapped to protein-coding transcripts and non-coding transcripts. Sites mapped to protein-coding transcripts were further divided based on the location on the transcript (coding sequence (CDS), exons, introns and untranslated regions (UTR)). Sites mapped to non-coding

transcripts were further divided into noncoding RNA (ncRNA), pseudogenes, 2000 bp upstream or downstream of an annotated genes or intergenic. **e** Distribution of editing within repetitive elements. **f** Two Sample Logo analysis displaying enriched and depleted nucleotides at edited sites. **g** Intersection analyses displaying overlap of edited sites in latent (left) and lytic (right) BCBL1 and BC-3 cells. **h** Sanger sequencing chromatograms of cDNA amplified from RNA and genomic DNA from BCBL1 and BC3 cells. The chromosomal coordinates (GRCh38.p13) for each editing site are listed below each chromatogram and the edited sites are indicated with a red arrow from the top. The nucleotides at each position are represented with a different color (Green = Adenosine, Black = Guanosine, Blue = Cytidine, Red = Thymidine). Source data are provided as a Source Data file.

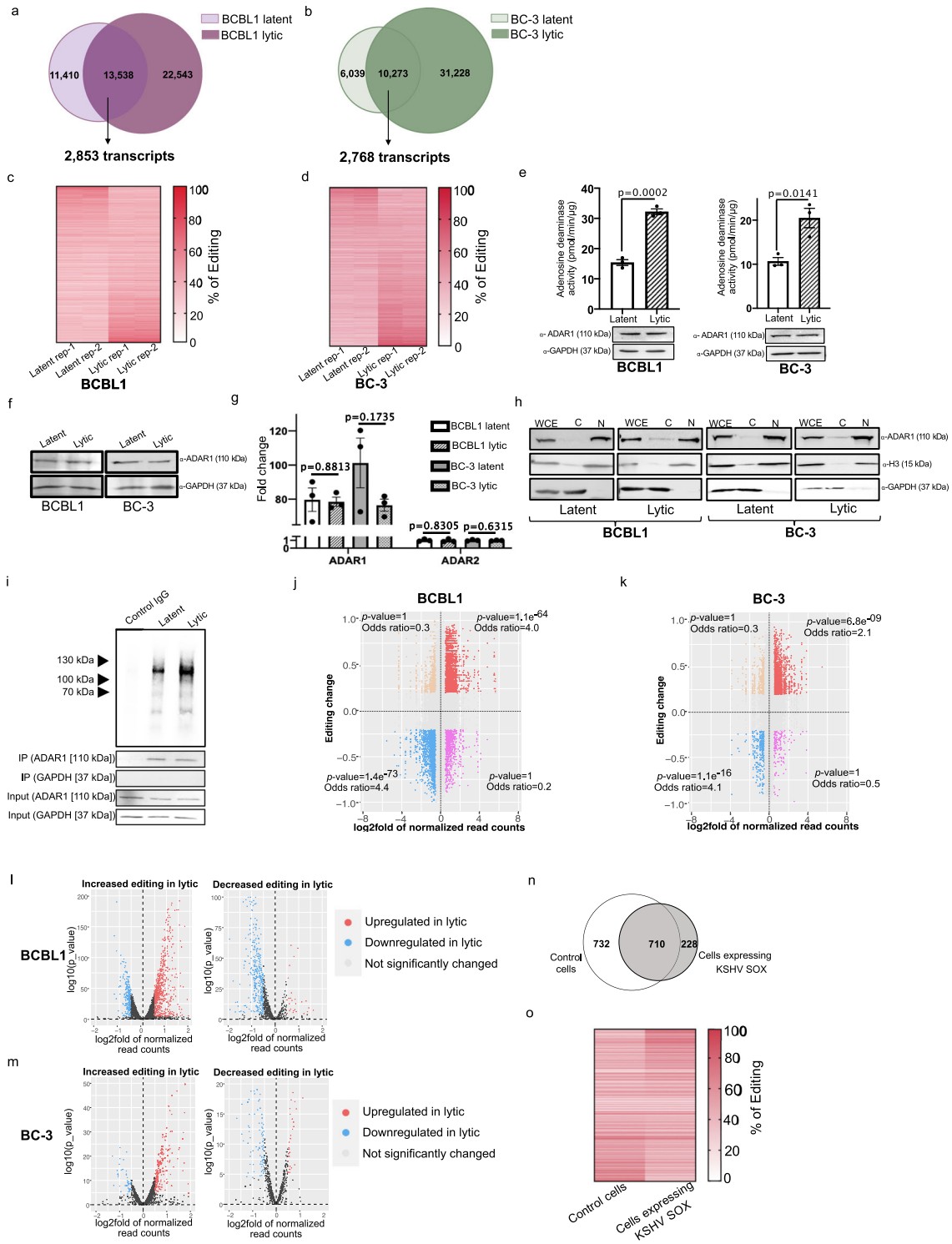

DNA and cDNA. In both BCBL1 and BC-3 cells we confirmed editing within the NOP14 and AJUBA transcripts, whereas the MAVS transcript exhibits BCBL1 specific editing events (Fig. 1h). Collectively, our transcriptome wide approach identified cell-specific editing and suggests remodeling of the editome upon lytic reactivation in PEL cells.

## Lytic reactivation increases adenosine deaminase activity

Our analyses indicate the host editome of lytic cells is 1.4-fold and 2.5-fold larger than latent BCBL1 and BC-3 cells, respectively. To further investigate this, we first sought to identify stage-specific editing events in the host transcriptome. We identified 11,410 and 22,543 sites that are specific to latent and lytic BCBL1 cells,

respectively (Fig. 2a). 13,538 sites are conserved between both cell states and these map to 2,853 transcripts. In BC-3 cells, 6,039 sites are latent-specific whereas 31,228 are lytic-specific (Fig. 2b). 10,273 sites mapped to 2768 transcripts are edited in both stages of KSHV infection in BC-3 cells. These analyses suggest that upon the latent to lytic transition there is a large increase in A-to-I editing of the host transcriptome. We also assessed the efficiency of editing at sites conserved in both latent and lytic cells. Strikingly, >70% of these sites exhibited increased editing upon lytic reactivation (Fig. 2c) and BC3 (Fig. 2d) cells. Thus, not only are more sites edited in the lytic transcriptome but editing stoichiometry is increased at conserved sites.

**Fig. 2 | Adenosine deaminase activity is increased during lytic reactivation.** Intersection analyses of latent and lytic host editomes in BCBL1 (**a**) and BC-3 (**b**) cells. **c**, **d** Heatmap of editing levels at sites conserved in (**a**) and (**b**). **e** Quantification of adenosine deaminase activity in lysates from BCBL1 and BC3 cells. Data are presented as mean values ± SEM, *N* = 3 biologically independent experiments. Statistical significance was calculated using a two-tailed Student's t-test. **f** Western blot analysis of indicated proteins in BCBL1 and BC-3 cells. **g** qRT-PCR quantification of *adar-1* and *adar-2* mRNAs. Data are presented as mean values ± SEM, *N* = 3 biologically independent experiments. Statistical significance was calculated using a two-tailed Student's t-test. **h** Western blot analysis of indicated proteins from subcellular fractionation. WCE, whole cell extract; C, cytoplasmic; N, nuclear. **i** Autoradiogram of ADAR1 CLIP from latent and lytic infected BCBL1 cells. Input lysates and 10% of the ADAR1 CLIP from BCBL1 cells were subjected to western blot against indicated proteins. *N* = 2 biologically independent experiments. **j**, **k** Correlation analysis between gene expression and editing activity in lytic BCBL1 (**j**) and BC-3 (**k**) cells. Significance of overlap was calculated using Fishers exact two-tailed t-test. Strength of association is represented by an odds ratio. **l**, **m** Volcano plots of transcripts that exhibit increased (left) and decreased (right) editing levels during lytic reactivation in BCBL1 (**l**) and BC-3 (**m**) cells. The P-adj values were calculated using the Wald test (default testing in Deseq2) and no further adjustments were made. Red and blue dots represent up- (P-adj <0.05, log2fold > 0.5) and downregulated (P-adj <0.05, log2 fold < −0.5) transcripts compared to latent cells. Gray dots represent genes that are not significant (P-adj > 0.05). **n** Venn diagram depicting number of edited sites in control HEK-293T and SOX overexpressing cells. **o** Heatmap of editing level at sites edited in both control and SOX overexpressing cells. Latent cells refer to the untreated PEL cells and lytic cells refer to the isolated GFP positive cells at 48 hpi. Source data are provided as a Source Data file.

Upon lytic reactivation we observe an increase in A-to-I edited sites as well as in editing efficiency at sites conserved in both latent and lytic cells. To further biochemically confirm these results, we quantified adenosine deaminase activity in whole extracts prepared from latent and lytic BCBL1 and BC-3 cells. Consistent with our previous results, lytic extracts from BCBL1 and BC-3 cells had significantly higher adenosine deaminase activity compared to latent control cells (Fig. 2e). The increase in activity is independent of expression changes of ADAR1, including the interferon inducible p150 isoform, as we detect no significant changes in its expression by western blot or qRT-PCR analyses (Fig. 2f, g). We did not observe the expression of ADAR2 by qRT-PCR or western blot (Fig. 2f, g).

As the levels of ADAR1 protein remained unchanged upon lytic reactivation, the increased editing levels could be caused by altered cellular localization of ADAR1. To investigate the localization of ADAR1 in lytic reactivation, we performed immunoblotting of nuclear and cytoplasmic fractions of latent and lytic BCBL1 and BC3 cells (Fig. 2h). Consistent with previous studies[13], our data indicated that ADAR1 is exclusively nuclear in latent BCBL1 and BC-3 cells. During reactivation, a minor portion of ADAR1 is detected in the cytoplasmic fraction, indicating relocation into the cytoplasm. Previous studies reported that ADAR1 is relocated to the cytoplasm when it is bound by dsRNA[37,38]. To test whether ADAR1 bound to more RNAs in lytic infection, we performed short wave-length crosslinking immunoprecipitation assays (CLIP) followed by $^{32}$P-radiolabeling (Fig. 2i). Briefly, lysates from UV crosslinked latent and lytic infected BCBL1 cells were digested with RNAse T1 and ADAR1 bound RNAs were immunoprecipitated with ADAR1 specific antibody. As a control, we performed an isotype control IgG immunoprecipitation from lytic extracts. Bound RNAs were radiolabeled with $^{32}$P and visualized by autoradiography. Consistent with our SAILOR and deaminase activity assays, the autoradiograph indicated that more RNA is bound to ADAR1 during the lytic cycle compared to latency. These data establish that upon the latent to lytic transition there is a significant increase in A-to-I editing.

Intrigued by the increase in editing in lytic cells we sought to investigate whether there was any correlation with gene expression. To test the correlation between editing and expression we plotted the log$_2$fold change in expression against the change in editing level at each site identified in BCBL1 (Fig. 2j) and BC-3 (Fig. 2k) cells upon the latent to lytic switch. Fishers exact t-test was performed to determine the statistical significance of the overlap. Additionally, an odds ratio was calculated to indicate the strength of association between expression and editing of a transcript. These analyses uncover a high degree of correlation between the expression change of an RNA and editing stoichiometry. This was further visualized by plotting the log$_2$fold change in expression of transcripts with increased editing and decreased editing separately (Fig. 2l and m). Consistent with correlation analysis, the majority of transcripts with increased editing are upregulated in expression upon lytic reactivation in BCBL1 (Fig. 2l left) and BC-3 (Fig. 2m left), whereas transcripts with decreased editing are downregulated in expression in lytic reactivation in BCBL1 (Fig. 2l right) and BC-3 (Fig. 2m right). Together, our data suggests that gene expression are correlated with the observed A-to-I editing changes during lytic reactivation.

KSHV encodes an endoribonuclease in ORF37 termed SOX that is responsible for wide-spread RNA decay during the lytic cycle. The destruction of cytoplasmic RNA via the viral endoribonuclease has been shown to promote the redistribution of RNA binding proteins (RBPs) from the cytoplasm to the nucleus[39,40]. We hypothesized that the SOX-dependent remodeling of the cytoplasmic RBP landscape may result in the unshielding of RNA leading to increased deamination activity. To test this, we identified A-to-I edited sites using SAILOR on previously published RNA-seq data from control and SOX over expressing HEK-293T cells[41]. 1442 sites are identified in control cells, whereas 938 sites are identified in SOX overexpressing cells (Fig. 2n). The smaller number of editing sites identified is likely a result of the reduced sequencing depth of the study. We compared the editing levels at the 710 sites that are identified in both control and SOX overexpressing HEK-293T cells (Supplementary Data 2). Nearly half of the sites (343 of 710) exhibited increased editing, while the remaining sites exhibited decreased editing upon SOX overexpression (Fig. 2o). Our analysis suggests that the overexpression of SOX does not impose a global change in A-to-I editing and that the increased editing during lytic replication is likely not a result of SOX-mediated unshielding. Collectively, our results demonstrate increased A-to-I editing during lytic reactivation in PEL cells and indicate that changes in gene expression correlate with remodeling of the host RNA editome.

### Expansion of the KSHV editome

To date, only one KSHV transcript, Kaposin A has been reported to be A-to-I edited[42–44]. To identify additional editing events we leveraged SAILOR and called A-to-I editing within the viral transcriptomes (Fig. 3a). SAILOR analysis identified 17 and 29 A-to-I editing sites within KSHV transcriptome with >99% confidence in BCBL1 and BC-3 (Supplementary Table S1) cells. To test whether the identified KSHV edited sites are conserved in PEL cells, we overlapped the sites identified in BCBL1 and BC-3 cells. Our data suggests the presence of conserved (6 in latent and 12 in lytic) as well as cell specific A-to-I editing within the KSHV transcriptome (Fig. 3b). Similar to the host editome, in both BCBL1 and BC-3 a greater number of edited sites were identified in the lytic cycle compared to latency.

To independently validate these results we selected five sites and performed Sanger sequencing. We validated the previously reported editing site within Kaposin A (chromosomal position: 117 809) along with previously unknown events within RTA and the KSHV miRNA cluster (Fig. 3c) in BCBL1 cells. The site within RTA is predicted to result in recoding of amino acid 378 from glutamic acid (E) to glycine (G), while the miRNA editing occurs with the pri-miRNA-K12-4 transcript. These data expand the editome of KSHV in PEL and further highlight the potential for A-to-I editing to influence the viral lifecycle.

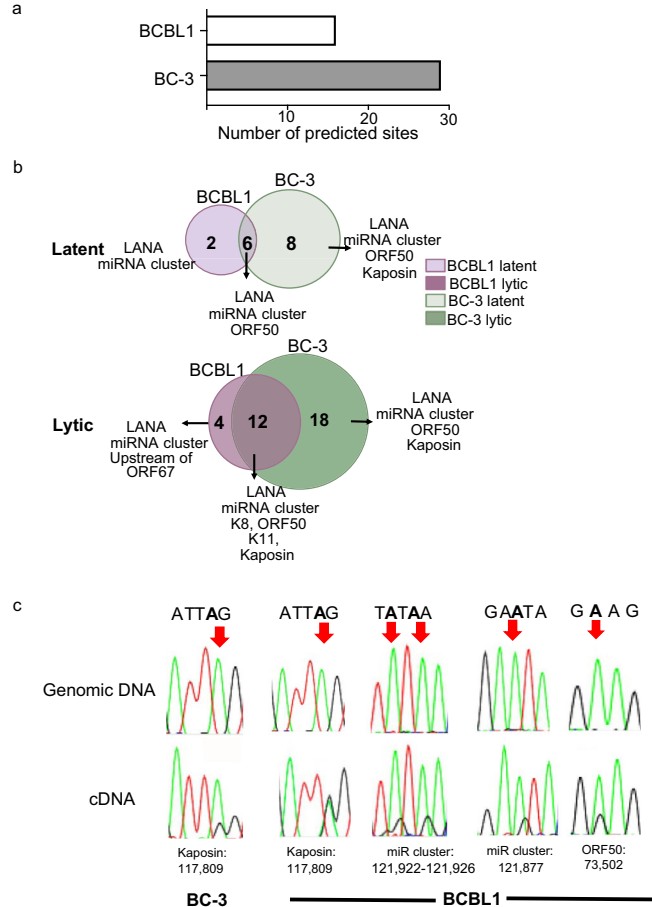

**Fig. 3 | The KSHV editome in PEL cells. a** Bar graph indicates the total number of SALIOR identified editing sites in BCBL1 and BC-3 cells. **b** Intersection analysis of latent (top) and lytic (bottom) BCBL1 and BC-3 cells. **c** Sanger sequencing chromatograms of cDNA amplified from RNA and genomic DNA from BCBL1 cells. The chromosomal coordinates (GQ994935.1) for each editing site are listed below the chromatogram. Edited sites are indicated with a red arrow from the top. The nucleotides at each position are represented with a different color (Green = Adenosine, Black = Guanosine, Blue = Cytidine, Red = Thymidine). Latent cells refer to the untreated PEL cells and lytic cells refer to the isolated GFP positive cells at 48 hpi. Source data are provided as a Source Data file.

## Editing within KSHV miRNA impacts miRNA biogenesis and target specificity

Given that the viral miRNAs play multiple roles in the viral lifecycle[45] we chose to further investigate the biological significance of their editing. Our analyses identified and confirmed three miRNA editing events that map to the pri-miRNA-K12-4 transcript. Two of the editing sites are within the lower stem region of pri-miRNA-K12-4 while the third site is within the seed region of mature miRNA-K12-4-3p (Fig. 4a). To determine whether A-to-I editing of pri-miRNA-K12-4 is conserved in other PEL lines we isolated small RNA and performed Sanger sequencing assays on four PEL lines, namely BC-1, BC-3, BC-5, and JSC-1 (Fig. 4b). BC-1, BC-5, and JSC-1 cells are co-infected with Epstein-Barr virus (EBV) while BC-3 cells are only infected with KSHV. We also assessed editing in iSLK-BAC16 cells, a common genetic model employed for KSHV in which the viral genome is encoded on a bacterial artificial chromosome (BAC). Our Sanger sequencing assays confirmed the presence of all three previously unknown editing events within pri-miRNA-K12-4 (Fig. 4b). To determine whether the editing events are mediated by ADAR1, we nucleofected BCBL1 cells with nontarget control or ADAR1-specific siRNA and evaluated editing 48 h post-siRNA depletion by Sanger sequencing. Depletion of ADAR1 resulted in a loss of A-to-I

editing within pri-miRNA-K12-4 (Fig. 4c). In addition, our data suggest that these pri-miRNA-K12-4 editing occurs within the nucleus (Supplementary Fig. 2) These data demonstrate the conservation of miRNA editing in PEL and conclude that it is ADAR1-dependent. Moreover, the presence of EBV in PEL lines, which is commonly observed in PEL, does not alter the ADAR1-dependent editing of pri-miRNA-K12-4.

Primary miRNA transcripts are bound by a heterotrimeric microprocessor, in which DROSHA acts as a "ruler" to cleave the stem region 11 nts from the base junction, while two DGCR8 proteins bind to the upper stem to enhance the efficiency and fidelity of DROSHA cleavage[46,47]. In addition to the structural elements, specific nucleotide sequences within the primary miRNA transcripts are reported to influence DROSHA cleavage[48,49]. The editing sites mapped to the lower stem region of pri-miRNA-K12-4 are 3 nts upstream of the DROSHA cleavage site. We hypothesized that proximity to the cleavage site may lead to alterations in miRNA biogenesis. To test this hypothesis, we leveraged an established KSHV miRNA expression vector[50] and mutated the corresponding adenosines within the pri-miRNA-K12-4 lower stem into guanosines, mimicking A-to-I editing, and quantified mature miRNA expression in HEK-293T cells by miRNA qRT-PCR (Fig. 4d left). We observed that mutation of the adenosine residues resulted in significantly less mature miRNA-K12-4-3p and −5p, suggesting editing decreases miRNA biogenesis (Fig. 4d right). As a control we quantified expression of miRNA-K12-9* derived from the transfected plasmid. We did not observe any significant reduction in expression of mature miRNA-K12-9*. In addition, northern blot analysis confirmed the reduction of mature miRNA-K12-4-3p when the lower stem was mutated to mimic A-to-I editing (Fig. 4e). To test the effect of editing on miR-K12-4 biogenesis in PEL we depleted ADAR1 in BCBL1 using siRNA knockdown (Fig. 4f). Consistent with our results in HEK-293T cells, depletion of ADAR1 resulted in an increase in mature miRNA-K12-4-3p and −5p by miRNA qRT-PCR (Fig. 4g). Together our results demonstrate that editing within the pri-miRNA-K12-4 lower stem impacts miRNA biogenesis.

miRNA target specificity is primarily determined by the sequence of the seed region. As we observe A-to-I editing within the seed sequence of miRNA-K12-4-3p we hypothesize that editing affects target specificity, effectively resulting in an expansion of RNAs that can be targeted (Fig. 4h). To test this, we first bioinformatically predicted targets of unedited and edited miRNA-K12-4-3p. Indeed, A-to-I editing is predicted to expand the repertoire of RNAs that can be targeted by miRNA-K12-4-3p. While 777 RNA are predicted to be targeted by the unedited miRNA, 277 RNAs are targeted by the edited miRNA. (Fig. 4i). In addition to the predicted increase of targets, A-to-I editing of miRNA-K12-4-3p expands the ontological associations of its targets. While unedited miRNA-K12-4-3p targets are ontologically enriched for terms involved in transcription, metabolic processes, and macromolecule biosynthesis, edited miRNA-K12-4-3p targets are enriched for cell growth and development related gene ontologies (Fig. 4j).

Our bioinformatic analyses suggest that A-to-I editing within the seed region of miR-K12-4-3p affects target specificity. Thus, we next sought to experimentally test this. We cloned the 3'UTR of six predicted targets downstream of a Renilla luciferase (RLuc) gene within a dual-luciferase construct and co-transfected it with WT or edited miRNA-K12-4-3p mimics. We selected targets that are regulated by both unedited and edited miRNA, targets that are preferentially regulated by unedited miRNA and targets that are preferentially regulated by edited miRNA. We quantified the expression of RLuc and normalized to the expression of Firefly luciferase (Fluc) (Fig. 4k). Luciferase assays confirmed that A-to-I editing impacts miR-K12-4-3p target recognition and the results match our predictions. Specifically, we identified targets that are unaffected by miRNA-K12-4-3p editing (WNT3 and TAB2), targets in which editing inhibits gene silencing (TPD52 and HECTD2), and another group in which the edited miRNA selectively represses gene expression (ROCK2 and EGR1) (Fig. 4k).

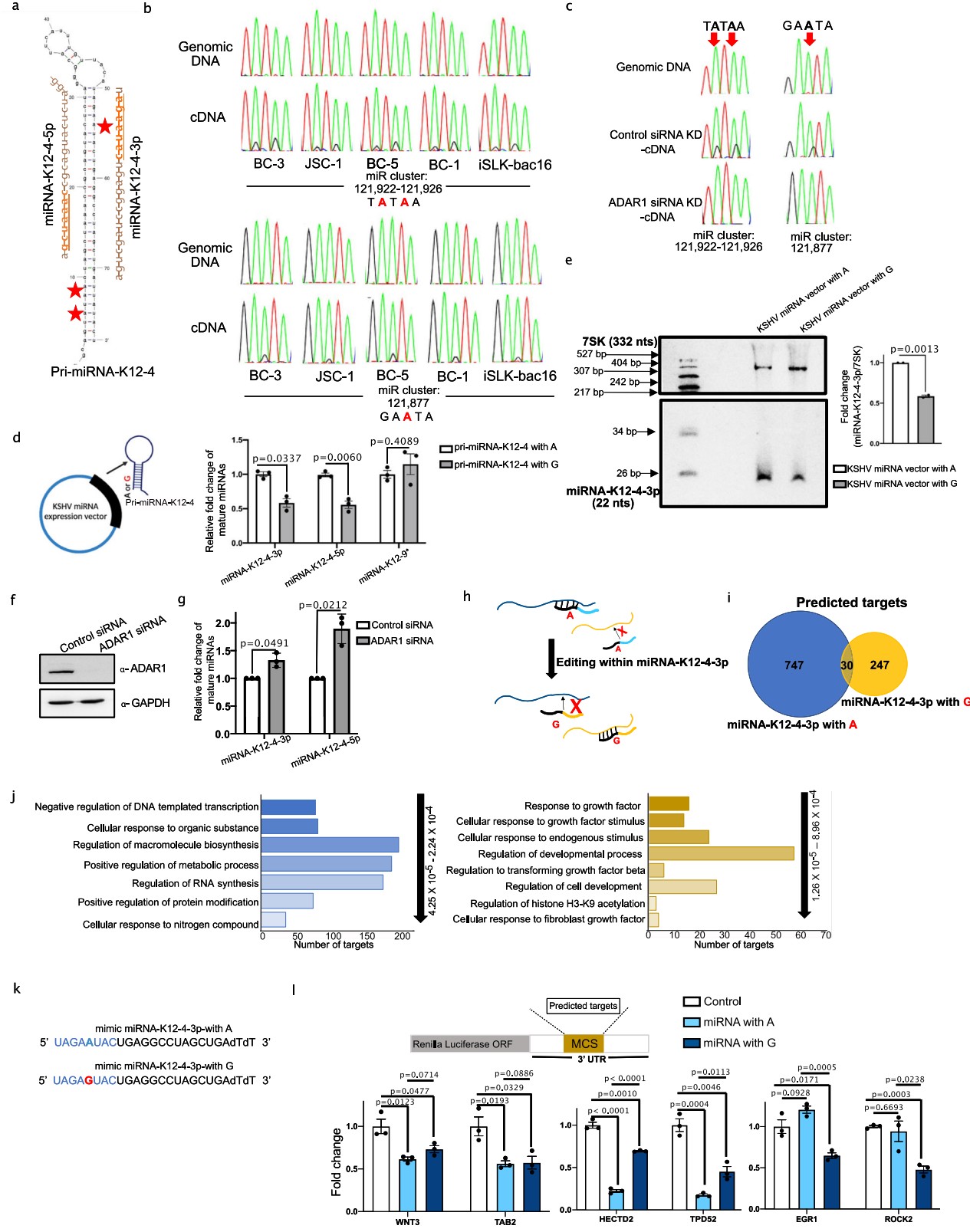

These data demonstrate that editing within the seed region of miRNA-K12-4-3p is capable of impacting target specificity and expands the potential of miR-K12-4-3p to regulate the host transcriptome.

## Editing within miRNA seed region impacts viral infection

KSHV miRNAs are expressed throughout the entire viral lifecycle. To investigate whether A-to-I editing within the seed sequence of miR-K12-4-3p affects the viral lifecycle we leveraged WT and miR-K12-4 deleted (Δ) iSLK-BAC16 cells (Fig. 5a). iSLK-BAC16 cells are a well-established system in which the KSHV genome is encoded on a BAC that constitutively expresses GFP. The presence of GFP can be used as a marker for cells infected with the virus. In addition, there is a doxycycline-inducible RTA encoded in the host genome that can facilitate lytic reactivation upon the introduction of doxycycline into the culture

**Fig. 4 | miRNA-K12-4 editing regulates miRNA biogenesis and target specificity.** **a** Predicted secondary structure of the pri-miRNA-K12-4 transcript. Mature miRNA sequences of miRNA-K12-4-5p and −3p are shown and seed sequences of both mature miRNAs are marked in bold and underlined. Red stars denote edited sites. **b** Sanger sequencing chromatograms of cDNA amplified from RNA and genomic DNA from BC-3, JSC-1, BC-1, BC-5 and iSLK-BAC16 cells. **c** Chromatograms of genomic DNA and cDNA amplified from RNA from ADAR1-siRNA or non-targeting control siRNA treated cells. In (**b**) and (**c**) the chromosomal coordinates for each edited site are listed below the chromatogram. Edited sites are indicated with a red arrow. The nucleotides at each position are represented with a different color (Green = Adenosine, Black = Guanosine, Blue = Cytidine, Red = Thymidine). **d** Cells were transfected with WT or miR-K12-4 mutant KSHV miRNA expression vector and mature KSHV miRNA expression was measured by Taqman. Results are shown relative to RNU48 and normalized to the siNeg control. Data are presented as mean values ± SEM, N = 3 biologically independent experiments. Statistical significance was calculated using a two-tailed Student's t-test. **e** Northern blot analysis of miRNA-K12-4-3p in total RNA isolated in (**d**). 7SK was used as loading control. Data

are presented as mean values ± SEM, N = 2 biologically independent experiments. Statistical significance was calculated using a two-tailed Student's t-test. **f** Lysates from ADAR1 or non-targeting control siRNA treated BCBL1 cells were subjected to immunoblotting with an ADAR1 and GAPDH antibody. **g** Quantification of miRNA-K12-4-5p and miRNA-K12-4-3p in RNA isolated from cells in (**f**) by Taqman assay. Results are shown relative to RNU48. Data are presented as mean values ± SEM, N = 3 biologically independent experiments. Statistical significance was calculated using a two-tailed Student's t-test. **h** Bioinformatic prediction of edited and unedited miR-K-4-3p targets using miRDB. **i**, gene ontology analysis of predicted targets in (**g**). **j** Sequence of miRNA mimics for unedited and edited miRNA-K12-4-3p. **k** 293T cells were transfected with luciferase reporter containing predicted 3'-UTR-targets along with one of the three miRNAs; unedited miRNA mimic, edited miRNA mimic, or non-targeting control miRNA. 24 h post-transfection luciferase activity was measured. Data are presented as mean values ± SEM, N = 3 biologically independent experiments. Statistical significance was calculated using a two-tailed Student's t-test. Source data are provided as a Source Data file.

media. We first sought to identify the impact of miR-K12-4 on lytic reactivation. Quantification of viral gene expression by qRT-PCR 48 h post-reactivation determined that the expression of ORF50, ORF57, and ORF45 was not impaired in miRNA-K12-4Δ virus relative to WT (Fig. 5b). Accordingly, transfection of an edited or unedited miR-K12-4-3p miRNA mimic did not significantly impact viral gene expression (Fig. 5b) or virion production upon reactivation (Fig. 5c).

To determine whether miR-K12-4 affects the production of infectious virions we performed supernatant transfer assays 72 h post-reactivation and quantified GFP+ cells by flow cytometry (Fig. 5d). Supernatants were transferred onto either HEK-293T cells or primary human umbilical vein endothelial cells (HUVECs). There was a striking reduction in GFP+ cells when media from miR-K12-4Δ virus was delivered. Moreover, transfection of a nontarget control or unedited miR-K12-4-3p into iSLK cells harboring the miR-K12-4Δ BAC did not rescue infectious virion production relative to the deletion virus. In contrast, transfection of edited miR-K12-4-3p resulted in a 10- and 6-fold increase of GFP+ cells relative to the deletion virus in HEK-293T and HUVECs respectively (Fig. 5d). These results demonstrate that while loss of miR-K12-4 does not affect lytic reactivation, virions produced from miR-K12-4Δ virus are significantly attenuated for infection. Moreover, complementation with edited miR-K12-4-3p significantly restores viral infection relative to control and unedited miR-K12-4-3p.

Given that virion production was not affected by the loss of miR-K12-4 we hypothesized that virion binding or entry were defective. To test this, we quantified viral attachment to HUVEC cells by incubating them with WT or miRNA-K12-4Δ virions for 90 min at 4 °C to allow attachment but prevent uptake, then measuring the relative level of attached virions by qPCR of the viral genome (Fig. 5e). Indeed, there was a significant decrease in virus binding to HUVECs when infected with miRNA-K12-4Δ virus relative to WT (Fig. 5f, left). We observed similar results when we quantified viral entry by incubating at 37 °C for 90 min, acid stripping, and measuring intracellular viral genomes by qPCR (Fig. 5f, right). We next tested whether virions produced from iSLK cells complemented with miR-K12-4-3p mimics restored virion binding and entry. While a nontarget control or unedited miRNA-K12-4-3p did not restore viral binding or entry, complementation with an edited mimic significantly enhanced both binding and entry (Fig. 5g). Collectively, these data demonstrate that miRNA-K12-4 is required for infectious virion production and that this phenotype can be partially rescued by complementation with edited miRNA-K12-4-3p. Thus, a dynamic A-to-I editome in PEL cells contributes to efficient KSHV infection (Fig. 5h).

## Discussion
The systematic profiling and characterization of RNA editing in species throughout evolution have revealed diverse cellular functions of A-to-I

editing. Accordingly, A-to-I RNA editing has emerged as an important contributor to the maintenance of cellular homeostasis in humans, and aberrant editing is associated with the onset of many diseases, including autoimmunity and tumorigenesis[51–53]. In contrast, the role of RNA editing in viral infection and pathogenesis is still emerging. Here, we describe a transcriptome-wide analysis of A-to-I editing during the KSHV lifecycle in PEL. Our study revealed that RNA editing is dynamic, and that lytic reactivation results in an expansion of both the host and viral editomes. Furthermore, we identify previously unknown RNA editing events within the KSHV miRNA cluster and demonstrate that they regulate viral miRNA biogenesis and target selection and are critical for efficient viral infection. Collectively, this work describes the PEL editome and reveals an RNA modification that is critical for the KSHV lifecycle.

During the latent to lytic transition we observe an increase in A-to-I editing. We and others have previously demonstrated that during lytic reactivation there is an increase in dsRNA[54,55] and it is likely that some of this becomes a substrate for RNA editing. Whether an increase in dsRNA is sufficient to increase editing is not known. Despite the increase in A-to-I editing we do not observe a significant increase in the expression of ADAR enzymes at the RNA or protein level. In addition, there is an interferon inducible p150 isoform of ADAR1, however, we and others do not observe its increased expression during lytic reactivation. Although, we do identify a high degree of correlation between the change in expression of an RNA and editing stoichiometry. We speculated increased RNA editing was a result of RNA unshielding mediated via the KSHV endoribonuclease SOX; however, we did not observe any alterations in the host editome in cells expressing SOX. ADAR activity can also be regulated via post-translational modifications. Along this line, stress-activated phosphorylation of ADAR1 by MKK6-p38-MSK1&2 MAP kinases facilitates its nuclear export where it can engage additional substrates[56]. In the context of lytic reactivation we do observe the relocalization of ADAR1 to the cytoplasm, raising the possibility that MKK6-p38-MSK1&2 MAP kinases contribute to the increase of A-to-I editing observed in lytic cells. In support of this, p38-MSK1&2 MAP kinases can be activated by multiple KSHV-encoded proteins[57]. Moreover, chemical inhibition or knockdown of p38-MSK1&2 MAP kinases reduce lytic reactivation[58–60]. Whether regulated ADAR1 relocalization and enhanced RNA editing contributes to this phenotype is intriguing and will be the subject of future studies.

Although our study focused on the functional and biological significance of viral miRNA editing our results raise the possibility of A-to-I dependent recoding of the host and viral proteome in PEL. Although human RNA editing has not been shown to be adaptive on a global scale, specific individual editing sites can have a functional effect. For example, editing within the coding region of the serotonin receptor 5-HT2C transcript regulates the sensitivity of the receptor to

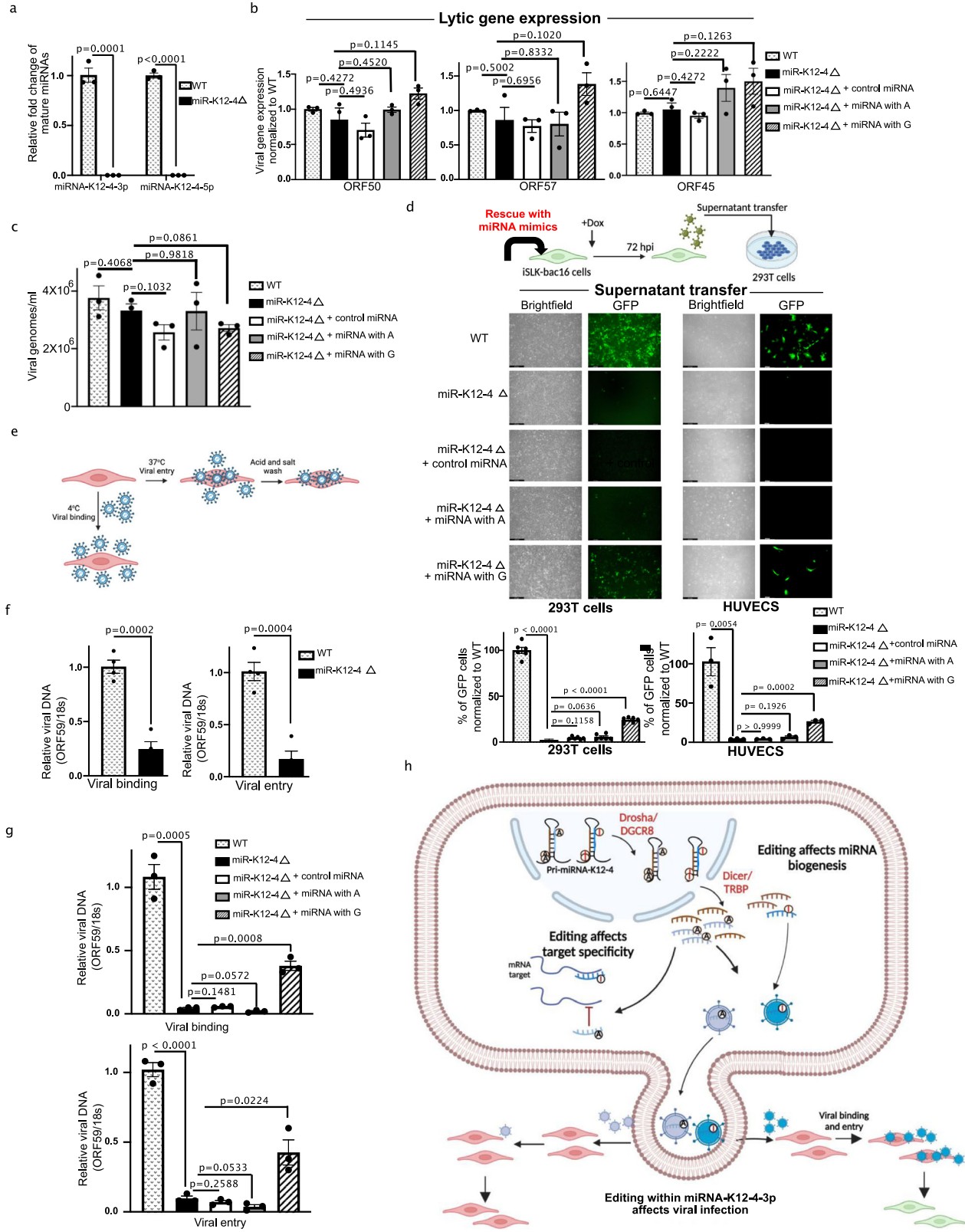

serotonin[61]. Moreover, a single site-specific editing event within glutamate receptor subunit GluR-2, altering a glutamine (Q) to an arginine (R) codon, is essential for normal brain function[62,63]. RNA editing within the potassium voltage-gated channel subfamily A member 1 (Kv1.1) is linked to temperature adaptation in octopus[64,65] and this suggest A-to-I editing can be adaptive and provide a survival advantage to an organism. Given the increase in host RNA editing upon lytic

reactivation we speculate the existence of adaptive editing events regulating host-virus interactions. Editing-dependent recoding of the KSHV genome may also be adaptive. Along this line, we did identify previously unknown RNA editing in exons of KSHV transcripts that are predicted to result in recoding, including within LANA, RTA, and vIRF2. Moreover, the editing event in RTA is within the DNA binding domain and results in the replacement of glutamic acid (E) with a glycine (G) at

**Fig. 5 | Editing within miRNA-K12-4-3p seed region affects viral infection.**
**a** Expression of miRNA-K12-4-5p and miRNA-K12-4-3p in WT and miRNA-K12-4 deleted iSLK-BAC16 cells was quantified by Taqman assay. Results are shown relative to RNU48. Data are presented as mean values ± SEM, N = 3 biologically independent experiments. Statistical significance was calculated using a two-tailed Student's t-test. **b** Quantification of viral gene expression by qRT-PCR 72 hpi in WT and miRNA-K12-4 deleted iSLK-bac16 cells with and without miRNA mimics. Data are presented as mean values ± SEM, N = 3 biologically independent experiments. Statistical significance was calculated using a two-tailed Student's t-test. **c** Quantification of viral DNA in supernatants from indicated viruses was measured by qPCR. Data are presented as mean values ± SEM, N = 3 biologically independent experiments. Statistical significance was calculated using a two-tailed Student's t-test. **d** Schematic of the supernatant transfer assays (top). GFP imaged 48 h post-transfer in HEK-293T (left) and HUVECs (right). Bar indicates 250 μm. Quantification of GFP cells performed using flow cytometry (bottom). Data are presented as mean values ± SEM, N = 6 biologically independent experiments for HEK-293T, and

N = 3 biologically independent experiments for HUVEC. Statistical significance was calculated using a two-tailed Student's t-test. **e**, Schematic showing depicting virus binding and entry assay. **f** qPCR used to quantify relative viral DNA levels. Data are presented as mean values ± SEM, N = 4 biologically independent experiments. Statistical significance was calculated using a two-tailed Student's t-test. **g** Supernatant transfer assays were performed as described in (**b**) using HUVECs and cells were incubated at either 4 °C (top) or 37 °C (bottom). qPCR used to quantify relative viral DNA levels from cells following incubation. Data are presented as mean values ± SEM, N = 3 biologically independent experiments. Statistical significance was calculated using a two-tailed Student's t-test. **h** The model depicting functions of A-to-I editing within pri-miRNA-K12-4. Editing at lower stem region of pri-miRNA-K12-4 results reduced amounts of mature miRNAs. Editing within the seed region of miR-K12-4-3p affects target specificity and viral infection. The model cartoon was created with Biorender. Source data are provided as a Source Data file.

residue 378. This results in the introduction of a small nonpolar amino acid in place of a negatively charged residue which may impact the ability of RTA to bind DNA and facilitate promoter transactivation.

The role of A-to-I editing in KSHV disease progression and therapeutic responsiveness warrants further investigation. Indeed, RNA editing has been demonstrated to influence oncogenic potential and responsiveness to therapy for several human malignancies. For example, exonic RNA editing and the subsequent recoding of S367G of AZIN1 (antizyme inhibitor 1) increases its oncogenic potential and is correlated with worse prognosis in several cancers, including hepatocellular carcinoma (HCC) and colorectal cancer (CRC)[66–68]. Moreover, cells expressing AZIN1 S367G are less sensitive to insulin-like growth factor type 1 receptor (IGF-1R) inhibition[69]. Editing of noncoding regions or noncoding RNAs can also promote cancer progression[70–72]. Intronic RNA editing of FAK (focal adhesion kinase) results in transcript stabilization and increases lung adenocarcinoma cell migration and invasion[73], while RNA editing of the tumor-suppressive microRNA pri-let-7d disrupts its biogenesis and is implicated in chronic myelogenous leukemia[74].

While ADAR1 has been suggested to be a KSHV proviral factor through the suppression of interferon responses in a RLR pathway dependent manner[44], the impact of RNA editing on KSHV-associated disease is not clear. KSHV Kaposin A (Kap A) is edited and its editing is increased by 10-fold upon KSHV reactivation[43]. Expression of Kap A has been reported to lead to transformation of rodent fibroblasts and an RNA editing-dependent recoding event, S38G, has been suggested to abrogate its transformative potential[43]. However, embedded with the Kap A transcript is miR-K12-10a and thus it was co-expressed in the previous study. More recent work has clarified that the miRNA is in fact responsible for transformation[75]; however, the role of editing on miRNA-mediated transformation was not investigated. Determining whether A-to-I editing affects miR-K12-10a-mediated transformation will be informative on mechanisms of KSHV-associated disease progression.

KSHV expresses 12 pre-miRNAs that are processed into 25 mature miRNAs from within latent locus[76]. In four PEL lines tested we identified the conservation of three A-to-I editing events within pre-miR-K12-4. Two editing events occur in proximity to the DROSHA cleavage site and reduce miRNA biogenesis, while the third is located within the seed sequence of miR-K12-4-3p and can influence target gene repression. Along this line, in our experimental analysis of six predicted miR-K12-4-3p targets we observe that RNA editing can both increase and reduce target miRNA-targeting efficiency in a transcript specific manner. Thus, RNA editing expands the targetome of miR-K12-4-3p. However, it is important to note that editing does not affect all targets.

KSHV is also the etiological agent of KS and Multicentric Castleman's disease (MCD) and we anticipate that the miRNA editing will also be conserved in these contexts. In support of the likelihood of miRNA editing in other contexts is our observation that miR-K12-4-3p is edited

in iSLK cells which are derived from a clear cell renal carcinoma, a type of kidney cancer. miRNAs can inhibit gene expression by at least two distinct mechanisms, transcript destabilization and inhibition of translation[77], thus future studies can leverage advanced transcriptomic and proteomic approaches to discern the effects of editing on target gene selection as well as mechanism of repression.

RNA modifications optimize RNA structure and function, providing an additional layer of post-transcriptional gene expression control. While we have established the landscape of viral and cellular RNA editing during KSHV latent and lytic replication in PEL, as well as identified functional A-to-I editing that impacts the viral lifecycle, there remains much to be investigated. Thus, these data will further serve as a rich resource for investigations into the KSHV lifecycle and viral disease and has the potential to uncover additional unappreciated regulatory strategies.

## Methods
### Cells and viruses
HEK293T (ATCC # CRL-3216), WT and miR-K12-4 deleted iSLK-BAC16 cells (a kind gift from Rolf Renne, University of Florida) were grown in Dulbecco's modified Eagle medium (DMEM; Invitrogen) supplemented with 10% fetal bovine serum (FBS; Invitrogen). HUVECS (ATCC # PCS-100-010) were grown in EGM-2 SingleQuots (Lonza #CC-4176) with the manufacturer provided supplements. TREx-BCBL1-RTA, BC3 (a kind gift from Britt Glaunsinger University of California, Berkeley) and the established PEL cell lines, TREx-BCBL1-PAN-GFP and iBC3-PAN-GFP cells were maintained in RPMI 1640 medium (Invitrogen) supplemented with 10% FBS (Invitrogen) and 2 mM L-glutamine (Invitrogen). All four of the PEL cells reported above have integrated doxycycline (Dox)-inducible version of the major viral transcription activator RTA. All cells were maintained with 100 U of penicillin/ml and 100 μg of streptomycin/ml (Invitrogen) at 37 °C under 5% CO2. Generation of TREx-BCBL1-PAN-GFP and iBC3-PAN-GFP cells: All lentivirus was prepared in 50–60% confluency HEK293T cells that were co-transfected with lentiviral construct, psPAX2 (Addgene), and pMD2.G (Addgene) using polyjet (SignaGen). 72 h post-transfection the supernatant was collected, adjusted to 8 μg/ml polybrene (Millipore), and target cells were spinfected at 1000 g for 1 h at room temperature. iBC3 cells were generated by subsequent lentiviral transduction with virions produced from pLenti CMV rtTA3 Hygro (Addgene) and pLenti-CMVtight-FL-HA-DEST-Blast (Addgene) harboring KSHV RTA. Cells were selected with 300 μg/ml hygromycin B (Invitrogen) and 5 μg/ml blasticidin (Invitrogen), respectively. cDNA constructs to express PAN-GFP were cloned into pLenti X1 Zeo/pTER and lentivirus was prepared has described above. Cells were selected for 4 weeks in media containing 100 μg/ml Zeocin (Invitrogen). The established TREx-BCBL1-PAN-GFP and iBC3-PAN-GFP are referred as BCBL1 and BC-3 in this manuscript. Cells were reactivated with 2 μg/ml of doxycycline (Dox; Fisher

Scientific) and GFP positive lytic cells were sorted using 5- laser FACS Aria III 48 hpi at VUMC Flow Cytometry Shared Resource Core (FCSR).

## Fluorescence in situ hybridization (FISH)

Approximately $5 \times 10^6$ TREx-BCBL1-PAN-GFP and BC3-PAN-GFP cells were collected from latent (untreated) and lytic (48 hpi with 2 µg/ml of Dox) infections and fixed in 4% (vol/vol) paraformaldehyde for 30 min at RT, washed with PBS-FISH buffer (1X PBS, 0.2 mg/ml RNase-free BSA) twice, and then permeabilized with 1× PBS containing 0.2% (vol/vol) Tween-20 for another 30 min at RT. The permeabilized cells were then hybridized with custom made Alexa-Fluor 647 end-labeled PAN anti-sense oligos (purchased from IDT, see Supplementary Table S2 for sequence) in HB 10% dx buffer (10% (wt/vol) dextran sulfate, 2× saline-sodium citrate (SSC), 10% (vol/vol) formamide, 1 mg/ml tRNA and 0.2 mg/ml BSA) at 37 °C overnight. Cells were washed twice with 500 µl of HBW buffer (2× SSC, 10% (vol/vol) formamide and 0.2 mg/ml RNase-free BSA), cells were analyzed on BD Canto II instrument. Data were analyzed with FlowJo X software (TreeStar). The gating strategy is shown in Supplementary Fig. 3.

## siRNA Knockdowns/miRNA mimic transfection

ADAR1 expression was depleted using siRNAs in TREx-BCBL1-PAN-GFP. TREx-BCBL1-PAN-GFP cells were nucleofected using Neon transfection system (Invitrogen) with 80 nM of ADAR1 siRNA pool and the non-targeting control pool siRNA twice at a 48 h interval at 1600 v, 10 ms pulse width, and three pulses. Twenty-four hours post nucleofection, cells were collected.

miRNA mimics were transfected into iSLK-BAC16 miRNA-K12-4 mutant cells using Lipofectamine RNAiMax (Invitrogen). iSLK-BAC16 miRNA-K12-4 mutant cells were transfected at 60–80% confluency with 80 nM of miRNA mimics. Twenty-four hours post-transfection cells were reactivated as described above and supernatant was collected from the cells to perform supernatant transfer assays.

## RNA isolation and qPCR

Total RNA was extracted using TRIzol (Invitrogen) and treated with DNase I (NEB). cDNA was synthesized with M-MLV Reverse Transcriptase (Promega) using random 9-mers (Integrated DNA Technologies). qPCR was performed using the PowerUP SYBR Green qPCR kit (Applied Biosystems 100029284) using appropriate primers on a QuantStudio 3 (Thermo Scientific).

For quantitation of KSHV-pri-miRNAs, RNA was isolated using the PureLink miRNA isolation kit (Invitrogen K1570-01). 1 ng of RNA was used for cDNA synthesis using TaqMan MicroRNA reverse transcription kit (Applied Biosystems 4366596) with miRNA specific RT primers (Applied Biosystems).

The mature KSHV miRNA were quantified using 0.67 µl of cDNA and microRNA primer probe sets (Applied Biosystems) (see Supplementary Table S2 for details), with TaqMan Universal PCR Master Mix, no AmpErase UNG (Applied Biosystems 4324018) according to manufacturer's instructions.

## Luciferase assays

HEK293T cells were transfected with psiCHECK2 or its derivatives harboring the target gene 3′-UTR fragments along with miRNA mimics using Lipofectamine 2000 (Invitrogen11668-027). Twenty-four hours post-transfection cells were collected and used to measure both renilla and firefly luciferase activity using dual-luciferase reporter assay system (Promega) on a GLOMAX 20/20 Luminometer (Promega).

## RNA-seq

RNA was isolated and DNAse I treated from latent and lytic cells as described above. Following DNAse I treatment, RNA was recovered by extraction with phenol:chloroform:isoamyl alcohol [25:24:1 (vol/vol)] followed by ethanol precipitation. Isolated RNA was subjected to

ribosomal RNA depletion using NEBNext rRNA depletion kits (Human/Mouse/Rat: NEB E6310) according to the manufacturer's instructions. The rRNA-depleted samples were used as the starting material to generate sequencing libraries using the NEBNext Ultra II directional RNA library prep Kit (NEB E7760) according to the manufacture recommendations. Libraries were then subjected to paired-end sequencing on a Nova-Seq with 150 cycles at the Vanderbilt Technologies for Advanced Genomics (VANTAGE).

## Bioinformatics analysis of RNA-seq reads

Raw reads quality in fastq files were assessed by FastQC (https://www.bioinformatics.babraham.ac.uk/projects/fastqc/). Raw reads were trimmed of adapters and aligned to the human genome (GRCh38.p13) and KSHV genome (GQ994935.1) using STAR (v2.7.3a) with the following parameters:

*[outFliterMultimap Nmax 1, outFilterScoreMinOverLread 0.66, outFilterMatchNminOverLread 0.66, outFlterMismatchNmax 20, outFilterMismatch NoverLmax 0.3–alignSJoverhangMin 8–alignSJDB overhangMin 1–alignIntronMin 20–alignIntronMax 1000000–alignMatesGapMax 1000000].*

Uniquely aligned reads were used as inputs to run featureCounts (v.1.5.2) to map the reads to gencode.v39 annotations using [-s 2] flag. For differential gene expression, DESeq2 (v1.18.1) was run with raw read counts obtained from featureCounts[78]. Transcripts that have a significant difference in gene expression (P-adj <0.05, using Benjamini–Hochberg correction) were differentially expressed. Gene Ontology analysis of the differentially expressed genes during lytic reactivation was obtained using http://geneontology.org/. To identify high confidence editing sites, uniquely aligned RNA reads were used as inputs for SAILOR[29]. Sites with a confidence of ≥0.99 were chosen for downstream analysis. Annotation of high-confidence sites was performed with a custom Python script using gencode.v39 annotations.

## Two Sample Logo analysis

The corresponding 5′ and 3′ neighbor nucleotides of the edited sites were retrieved using BEDTools (Quinlan and Hall 2010). Five-nucleotide stretches (centered on the edited adenosine) were randomly picked from human genome and were used as the background to determine the overrepresented and underrepresented nucleotides around the edited sites using Two Sample Logo (http://twosamplelogo.org).

## Sanger sequencing validation of edited sites

RNA was extracted from the latent and lytic infected cells as described above and then reverse transcribed with gene specific primers (Supplementary Table S2) using Superscript IV (Invitrogen) followed by PCR with HiFi HotStart (KAPA biosystems #07958897001). For each sample, negative controls without Superscript IV were conducted to assess the DNA contamination.

## Western blotting

Whole-cell lysates were prepared with lysis buffer (50 mM Tris [pH 7.6], 150 mM NaCl, 0.5% NP-40, protease inhibitor cocktail [Thermo scientific # 1861279,100X]) and quantified by Bradford assay (BioRad). Equal amounts of extracts were resolved by SDS-PAGE, electrotransferred to PVDF membrane (Millipore), and probed for the indicated proteins. Antibodies: ADAR1 (Cell signaling technology, #81284 s, 1:1000), Histone H3 (Millipore, #05-928, 1:1000) and GAPDH (Proteintech, 60004-1-Ig, 1:5000). Primary antibodies were followed by Alexa-Fluor 680-conjugated secondary antibodies (Life Technologies, goat anti-rabbit #A27042, goat anti-mouse #A28183, 1:10,000) and visualized by an iBright FL1500 system (Thermofisher).

## Nuclear cytoplasmic fractionation

Approximately $5 \times 10^6$ cells were washed with cold PBS and centrifuged at $200 \times g$ for 5 min at 4 °C. The pellet was re-suspended in 1 ml cold

PBS containing 0.1% NP-40 and a portion of this whole cell extract (WCE) was saved. The remaining WCE was subjected to pop-spin at 4 °C and the supernatant containing the cytoplasmic fraction and the pellet containing nuclear fraction were separated. Pellet was washed with 1 ml cold PBS containing 0.1% NP-40 to remove the cytoplasmic contaminants and re-suspended in 100 µl of cold PBS containing 0.1% NP-40. Equal volumes of WCE, cytoplasmic and nuclear fractions from each sample were subjected to Western blotting as described above.

### Crosslinking immunoprecipitation (CLIP) assays

Approximately $5 \times 10^6$ cells were washed with cold PBS and subjected to UV irradiation at $0.3 \, J/cm^2$ twice in a 254 nm crosslinker. Cells were detached from the crosslinked plate using cell scraper and pelleted at $1000 \times g$ for 5 min. Cell pellets were lysed in iCLIP buffer (50 mM Tris HCl pH 7.4, 100 mM NaCl, 1% NP-40, 0.1% SDS, 0.5% Sodium deoxycholate, protease inhibitor cocktail [Thermo scientific # 1861279,100X]) on a rotator for 20 min at 4 °C. Cell lysates were centrifuged to pellet the cellular debris and supernatant was transferred to a new tube. RNAse T1 (Thermo scientific #EN0542) was added to the supernatant at a final concentration of 1 U/µl and incubated at 22 °C for 15 min shaking at 1000 rpm. Following incubation, the reaction was placed on ice for 5 min. anti-ADAR1 coated Protein G magnetic beads (Biorad #161-4023) were added to the lysate and incubated for 2 h at 4 °C. The beads were washed with IP wash buffer (20 mM Tris HCl pH 7.4, 10 mM MgCl$_2$, 0.2% Tween-20), RNase T1 was added to the beads to a final concentration of 100 U/µl and incubated at 22 °C for 15 min in a shaker (1000 rpm). Following the RNase T1 digestion, beads were again washed with high salt wash buffer (50 mM Tris HCl pH 7.4, 1 M NaCl, 1% NP-40, 0.1% SDS, 0.5% Sodium deoxycholate, 1 mM EDTA) and 1/10 of the beads were saved for Western blotting. The beads were resuspended in dephosphorylation buffer (10× buffer: 50 mM Tris HCl pH 7.9, 100 mM NaCl, 10 mM MgCl$_2$, 1 mM DTT) containing CIAP (Promega M182A) at a final concentration of 0.5 U/µl and incubated at 37ºC for 10 min with shaking at 1000 rpm. The beads were washed with PNK buffer (50 mM Tris HCl pH 7.4, 50 mM NaCl, 10 mM MgCl$_2$) 2X before the phosphorylation reaction with PNK (Thermo scientific EK0032) and $^{32}$P γ-ATP (Perkin Elmer). Finally, beads were washed with NT2 (50 mM Tris HCl pH 7.4, 150 mM NaCl, 1 mM MgCl$_2$, 0.005% NP-40) buffer 2×, heated to 95ºC and resolved on SDS-PAGE gel. Gels were exposed to phosphorimager screens overnight and autoradiographed.

### Deamination activity assay

Approximately $5 \times 10^6$ latent and lytic infected cells were collected and washed with PBS. The cellular adenosine deaminase activity was measured by the Adenosine Deaminase (ADA) Activity Assay Kit (Abcam, ab204695) according to the manufacturer's instruction.

### Supernatant transfer assay

For supernatant transfer, WT and miRNA-K12-4 deletion iSLK-BAC16 cells were reactivated with Dox (2 µg/ml) for 72 h. For the complementation experiments with miRNA mimics, 80 nM microRNA mimics were transfected with Lipofectamine RNAiMax (Invitrogen) and the cells were reactivated 24 h post-transfection The supernatant was collected and 25% of the supernatant was supplemented with 8 µg/ml polybrene and 40% polyethylene glycol (PEG) 8000. HEK-293T and HUVECs cells were spinfected with the supernatant at $1000 \times g$ for 1 h and 30 min respectively at room temperature (RT). The infection media was replaced with fresh media and incubated for 48 h prior to imaging for GFP positive cells and flow cytometry analysis.

### Northern blot Analysis

Thirty micrograms of total RNA per sample was separated on 8% polyacrylamide–7 M urea gels and electrotransferred at 4 °C to Amersham Hybond-N+ membranes in 0.5× TBE buffer for 16 h at 15 V. Membranes were probed overnight using 32P-end labeled probes (see

supplementary Table S2) overnight at 55 °C. Blots were washed three times in 0.1× SSC for 10 min each before exposed to phosphoimager screens overnight.

### Isolation and quantification of recombinant virus from iSLK-BAC16 cells

WT and miRNA-K12-4 deleted iSLK-BAC16 cells were induced with Dox (2 µg/ml) and Sodium butyrate (1 mM) for 96 h. The supernatant was subjected to ultracentrifugation through a 20% sucrose cushion at $100,000 \times g$ for 1 h. Virus pellets were dissolved in 500 µl of using serum free DMEM, aliquoted and stored at −80 °C. Viral DNA from 50 µL of virus aliquots was isolated using genomic DNA isolation kit (Promega #A1120) according to the manufacturer's instructions. Virus DNA was resuspended in 20 µL of H$_2$O and 2 µL was used per qPCR reaction. Real-time qPCR was performed using serial dilutions of isolated KSHV BAC16 as standards with primers specific for ORF59. The viral genome copy number was determined by comparing viral DNA to the BAC16 standard curve.

### Viral binding and entry assays

Viral genome numbers are quantified using qPCR as described above. HUVECs cells were incubated with virus (MOI = 3 genome equivalents) supplemented with 8 µg/ml polybrene and 40% polyethylene glycol (PEG) 8000 at $1000 \times g$ 4 °C (to measure binding) or 37 °C (to measure entry). The cells were incubated at either 4 °C or 37 °C for 90 min. For the binding assay, cells were then washed 2X with ice-cold PBS, scraped. Viral DNA was isolated using genomic DNA isolation kit (Promega #A1120). For the entry assay, cells were washed 2× with PBS, and 0.5 ml of citric acid [135 mM NaCl, 10 mM KCl, 40 mM citric acid, pH 3] was added for 5 min at RT to strip off remaining cell surface-bound virus. Cells were washed 2× more with PBS, scraped, and viral DNA was isolated. Relative genome levels were quantified by qPCR with ORF59 specific primers.

### Reporting summary

Further information on research design is available in the Nature Portfolio Reporting Summary linked to this article.

## Data availability

Sequencing data from this study have been deposited in SRA under project number PRJNA875094 (GEO under accession number GSE212350). Sequencing data (GEO under accession number GSE128866) from a previous study have been used to analyze the effect of SOX in RNA editing. High throughput sequencing reads were aligned to the human reference genome (gencode GRCh38.p13, https://www.ncbi.nlm.nih.gov/assembly/GCF_000001405.39/) and KSHV genome (GQ994935.1, https://www.ncbi.nlm.nih.gov/nuccore/GQ994935.1). All the transcripts were summarized to biotypes annotated with GENCODE database (gencode.v39, ftp://ftp.ebi.ac.uk/pub/databases/gencode/Gencode_human/ release_39/gencode.v39.chr_patch_hapl_scaff.annotation.gtf.gz). Gene ontology analysis on the differentially expressed genes were performed using http://geneontology.org/. Source data are provided with this paper. A reporting summary for this article is available as a Supplementary file. Source data are provided with this paper.

## Code availability

Annotation of high-confidence sites was performed with a custom Python script using gencode.v39 annotations. This custom python script and the R scripts that were used to analyze differential gene expression are deposited in Github (https://github.com/yexiang2046/AIediting).

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

## Acknowledgements

We would like to thank members of the Karijolich lab for insightful discussion. We thank Britt Glaunsinger (University of California, Berkeley) Jae Jung (Lerner Research Institute, Cleveland Clinic), Rolf Renne (University of Florida) and Joseph Ziegelbauer (National Cancer Institute, NIH, Bethesda) for providing reagents. The Karijolich laboratory was supported by startup funds from Vanderbilt University Medical Center and National Institutes of Health (NIH) grants R01AI141448 (to J.K.), R01CA250051 (to J.K.), and an American Cancer Society Research Scholar Award RSG MPC – 133907 (to J.K.). J.K. is a Pew Biomedical Scholar. The cartoons in Fig. 5d, e, h were created with Biorender.com.

## Author contributions

J.K. and S.R. conceived and designed the experiments. S.R., X.Y., W.D., and A.R. performed the experiments. The data was analyzed and the manuscript was written by S.R. and J.K.

## Competing interests

The authors declare no competing interests.
