## [Peer Review File · Nature Communications]

Reviewers' Comments:

Reviewer #1:

Remarks to the Author:

Summary

The manuscript by Rajendren et al. provides the first global analysis of the A-to-I editome in PEL cells and KSHV transcriptome during KSHV latent and lytic replication. Using modified PEL cell lines, the authors collected latent and lytic GFP+ cells for deep RNA sequencing analysis to identify A-to-I edited sites using the SAILOR algorithm and further validated editing of specific sites by comparing the cDNA with genomic DNA through Sanger sequencing. Excitingly, the authors identified conserved and stage-specific A-to-I editing sites in the transcriptome of host BCBL1 and BC-3 cells and KSHV during latent and lytic reactivation. The authors further identified three previously unknown ADAR1-mediated conserved editing sites of the KSHV pri-miRNA-K12-4 transcript, two in the lower stem region and one within the seed region of mature miRNA-K12-4-3p, which affect (1) miRNA biogenesis, (2) target specificity, and (3) viral transmission. Using an established KSHV miRNA vector, the authors mutated the adenosines into guanosines within the lower stem region of pri-miRNA-K12-4 and showed a (1) significant decrease in mature miR-K12-4-3p and -5p not observed upon ADAR1 siRNA-mediated depletion. Using bioinformatics tools and luciferase assays, the authors showed that (2) editing of the seed sequence of miRNA-K12-4-3p expanded target recognition. Using the iSLK-BAC16 KSHV infection cell model with rescued edited miR-K12-4, the authors showed that (3) editing at the seed region increases infectious virion production. These findings shed light into the functional and biological significance of A-to-I editing in KSHV latent and lytic replication in PEL cells, opening the door for future applications. However, main concerns are the lack of ADAR1 isoform detection that may impact data interpretation, the importance of seed region editing of mature miRNA-K12-4 for KSHV transmission, lack of in-depth discussion of host and KSHV transcriptome remodeling during viral stages, and lack of percent editing in the Sanger trace to showcase the population of RNA edited. Additional experimentation is required to support the reported findings.

Major critics

- The authors do not take into consideration the reported isoforms of ADAR1. Based on reported literature, it is likely that in latent infection, p110 is the main isoform which is predominantly found in the nucleus and correlates with Fig 2h ADAR1 nuclear localization. Upon lytic reactivation, the IFN-inducible p150 isoform may be expressed which localizes into the cytosol and has been reported to edit more substrates and more sites than p110.1 The increased editing activity the authors observed may not only be due to increased gene expression but also induction of p150 isoform. The authors need to address both isoforms and show their expression in respective western blots. In lines 366-370, the authors discussed the slight localization of ADAR1 (i.e., p110) known to shuttle between the nucleus and cytosol upon stress (Fig 2h), but neglect the major cytosolic isoform, p150.
- In Fig 5d, the authors show that editing within the miRNA-K12-4-3p seed region impacts viral transmission. However, whether this editing is required by KSHV for transmission still needs to be further addressed. The authors previously describe how the pri-miRNA K12-4-3p is edited in the lower stem loop and this significantly decreases mature miRNA K12-4-3p expression in an anti-viral fashion. Following miRNA processing, primary miRNA editing should occur before the editing of the mature miRNA seed region, which appears to be pro-viral. The authors could include wt and edited pri-miRNA-K12-4 in their experiments to assess whether the editing of the cleaved miRNA-K12-4 is truly required for viral transmission and the findings are not artificial. On this note, the authors should also address which type of editing is preferred within the primary vs mature miRNA transcript in either latent or lytic stage of replication. Since ADAR1 p110 is in the nucleus, it is likely the predominant editor of pri-miRNA-K12-4, while p150 may be the predominant editor of miRNA-K12-4.
- In order to discreetly analyze the impact of A-to-I RNA editing in KSHV reactivation (latent vs lytic cycle), it would be important to include the comparison of global A-I editing sites, efficiency, and its transcriptional implication between cells that were induced for lytic cycle but were GFP- cells (bystander cells) within the same population. This analysis might provide insights of the mechanism of "spontaneous" lytic reactivation.

- The percentage of editing (peak height of G/(G+A)*100) in sanger sequencing could give a more accurate representation of the population of mRNA edited. This is particularly necessary for the miRNA editing observed (Fig 5d), where there seems to be lower editing compared to other mRNA regions (Fig 1h), which the authors need to address.
- The authors should focus on the novelty of their global analysis finding, further discussing specific host and KSHV transcript remodeling between latency vs lytic reactivation in addition to miRNA K12-3-4p editing sites and less emphasis on increased numeric values upon lytic reactivation. Gandy et al had previously reported that ADAR1 A-to-I editing controls the function of the Kaposin A/mi-K10 portion of the K12 transcript in PEL cells and that editing increased 10-fold by activation of lytic replication.² On this note, Zhang et al. also showed that ADAR1 facilitates KSHV lytic reactivation by modulating the RLR-dependent signaling pathway.³ The field also knows that ADAR1 preferentially edits intronic and untranslated regions mainly in host Alu elements.^{1,4}
- Lines 371-385, the authors discuss the potential of editing in protein coding regions. However, this study showcases how most host editing occurred in intronic (>60%) and untranslated regions (~30%) and focus on non-coding KSHV miRNA. The authors showed Kaposin and ORF50 (Fig 3c) editing in BCBL-1 cells, but this requires further follow up studies to prove adaptive RNA editing. On this note, unlike host analyses (Fig 1d), the authors did not detail the percent editing in different types of KSHV RNA, which could strengthen their interpretations and add to their findings.

Minor critics

- Line 54, as per the Wiley citation, it is small nucleolar RNA (snoRNA) not snRNA and the authors mention they promote 2'O-methylation of adenosine which impairs editing. As per the Nishikura citation, the authors report that adar1p110 accumulates in the nucleolus proposed to binding to rRNA or to small nucleolar RNA.
- Line 59, ADAR1 KO mice not only have reduced adar1 editing, but the phenotype is also lethal, mainly through MDA5-mediated mechanism
- Line 358, it has been reported greater Z-RNA repertoire allow p150 rather than p110 to bind to more RNA and have increased activity.⁵
- Line 380, please further describe and cite what is meant by adaptive RNA editing
- Line 408, authors mention five PEL lines tested for conserved editing sites in pri-miR-K12-4 yet only four PEL lines are mentioned in line 244: BC-1, BC-3, BC-5, and JSC-1.
- F1 b, may be unnecessary
- F1 d, donut charts are simple to follow but the order and arrows into protein coding and non-coding regions needs to be simpler to follow. They also have a subset between latent and lytic which makes it more complicated.
- F1 e-f, can be supplemental
- Fig 1h, there needs to be an average percent editing based on peak height
- Fig 1b, the text (lines 103-104) describes how we can compare cDNA sequence and reference genomic sequence to identify A-to-I RNA editing sites. However, figure 1B shows how adenosine is converted to inosine at the RNA level. The text and figure do not match well.
- Fig 1d, may need to check the % of distribution of non-coding transcription again. The number of genes expressed in BCBL1 and BC-3 were different, but the % of distribution of the latent and lytic in the two cells showed the same percentages.
- Fig3e, y axis is ADAR activity, not ADA
- Fig 3c, were BC-3 cells used to see the editing sites within Kaposin A?
- Fig 4e, what should be the size of these bands? The size of the KSHV miRNA vector with G seems to be lower than the KSHV miRNA vector with A
- Fig 4b, Two "BC-3" texts overlap each other
- Fig 4e, Extra box presents in the key section
- Fig 2f, MW are missing on the western blot
- Fig 4d and Fig 5c, need improved replicated for less variation and accurate statistics
- Fig S1c, need labels for BCBL1 and BC-3 cell type

References

1. Sun, T. et al. Decoupling expression and editing preferences of ADAR1 p150 and p110 isoforms.

Proc. Natl. Acad. Sci. U. S. A. 118, (2021).

2. Gandy, S. Z. et al. RNA Editing of the Human Herpesvirus 8 Kaposin Transcript Eliminates Its Transforming Activity and Is Induced during Lytic Replication. *J. Virol.* 81, 13544–13551 (2007).

3. Zhang, H., Ni, G. & Damania, B. ADAR1 Facilitates KSHV Lytic Reactivation by Modulating the RLR-Dependent Signaling Pathway. *Cell Rep.* 31, (2020).

4. Bahn, J. H. et al. Genomic analysis of ADAR1 binding and its involvement in multiple RNA processing pathways. *Nat. Commun.* 2015 6:1–13 (2015).

5. de Reuver, R. et al. ADAR1 interaction with Z-RNA promotes editing of endogenous double-stranded RNA and prevents MDA5-dependent immune activation. *Cell Rep.* 36, (2021).

Reviewer #2:

Remarks to the Author:

In this paper, the authors explore the roles of A-to-I editing in KSHV-infected cells. They first catalog A-to-I editing within latent and lytic cells and show that the A-to-I editome changes between phases. They examine both host and viral gene targets and then delve more deeply into the effects of KSHV miRNA editing on processing and function. Overall, the data are clearly presented and interpreted appropriately. The findings impact both the KSHV and RNA editing fields, so it has a broad audience. I have only minor comments for improvement of the manuscript:

1. Line 97: "We collected latent and lytic GFP + cells". This reads as if latent cells are GFP positive. Please clarify this sentence in the text.
2. Please label the two graphs in Figure S1c on the diagrams (BCBL1 and BC3). Similarly, label Fig 2l/m on graphs.
3. X-axes are often labeled log₂fold. Log₂-fold of what? Some quantity should be given in the label.
4. Color code key should be given for the Sanger sequencing traces.
5. In Fig 4a, please graphically depict the sequence of the mature miRNAs and seed sequences on the pri-miRNA. It would be helpful for readers to know where these features are to compare them to the sites of editing.
6. In Fig 4d the miRNA is labeled K4 instead of K12-4.
7. For the data presented in Fig 4k, they state that they "cloned the 3UTR of six predicted targets...". How are they defining "predicted" here? Does this mean predicted to be targeted by both, edited, or unedited miRNAs? If they are specific, do the results correlate with predictions?
8. The authors correctly state that the data in Fig 5c for the complementation with "miRNA with G" are not statistically significant. However, 2 replicates are quite a bit higher than the third and the lower mean and higher error is due to the third one replicate. This may point to biological relevance, particularly in light of the results in Fig 4d. How confident are they that the lytic replication is not better in these cells? This should perhaps be noted.
9. Overall, the labels on figures tend to be too small.

Reviewer #3:

Remarks to the Author:

This manuscript reports the A to I editing patterns of cells infected with KSHV virus, and in either the latent or reactivating state of infection. They demonstrate that there is a substantial increase in A-I editing of the KSHV genome and of the cellular genome in cells undergoing lytic reactivation, and that the A-I editing pattern in these 2 states of infection has some overlap, but also has very clear distinctions in the A-I edited sites. These findings are exciting and novel, and provide insights into an important source of RNA editing during infection, with potential for biological outcomes during infection. They go on to show that one of the edited sites in a viral miRNA results in a functional change in the miRNA, such that the result is a significant decrease in virus lytic reactivation. These new insights are strong and are important to the fields of RNA biology and to virus-host interactions.

The major conclusions and large datasets reported in this manuscript (above) are sound, yet there

are some secondary conclusions and details that are not strong enough to support the conclusions. Particularly, the authors have not demonstrated that modification of the miRNA-K-12-4 sequence results in downstream defects in virus binding and entry.

Major concerns:

1) Several experiments use transfected/modified cells that are not well-described and for which there is no validation data provided. In nearly every case, the pedigree and timing of the cells should be clarified for the reader, potentially by timeline schematics or tables.

a) the pPAN reporter cells are very briefly described, and are essential to the study. Data should demonstrate their inducibility relative to parent lines and their GFP status with and without induction. Evidence that the sorts result in pure populations of latent vs lytic reactivating cells could be further supported by the RNA Seq files. Flow cytometry data should include stained and unstained samples, controls, live/dead discrimination, gating strategies, and comparison of pre-sort and post-sorted cells. Clarification on whether 'latent cells' refer to untreated cells or cells sorted and selected as GFP-negative. The same goes for the PAN fish flow staining, PAN anti-sense oligos should be listed, and methods are unclear on which Alexafluor is used in experiments shown.

b) the iSLK and BCBL-TRex cell transfection schemes should be clarified. There are a number of miRNA and mimics and each could be made more plain for the reader. Text and Methods differ in transfection methods.

2) While the manuscript does a very convincing demonstration of A-I editing distinctions and shows quite well that A-I editing of a viral miRNA compromised virus lytic reactivation, there are subsequent studies of the resulting virus that are not well supported. In Figure 5d, the authors show that miRNA-K12-4 mimics with sequences matching the edited or unedited sequences result in striking differences in virus reactivation with the result of significantly decreased virus production. However, in Figure 5f-h, the data shown is taken to mean that the resulting virus from this supernatant is itself defective in subsequent binding and entry. If this were to be proven, it would require inoculation of equivalent amounts of virus for analysis of binding and entry. This is a significant burden of proof, and could be quite difficult given the very poor virus production shown in Figure 5c/d. Methods state that these cells were infected with an MOI=3 particles, but no particle measurements can be made without EM or Virocyte, so it is unclear what is meant here. Viral DNA is reported as relative to WT rather than relative to input, and no limit of detection or background level is provided. As is presented, there is not convincing data shown to refute the reasonable conclusion that reduced virus production results in reduced virus available for binding and entry.

3) The schematic refers to virus infection as transmission, however, virus transmission is the process by which viruses spread between hosts.

MINOR points:

1) Figure 2h, there is some H3 signal in the cytoplasmic fractions. Comment on whether this relates to virus infection and nuclear membrane integrity?

2) Figure 2j/k: comment on relatively rare events beyond 2-fold changed?

3) Figure 4e has an unlabeled white box below "KSHV miRNA vectore with G", and Figure 4e has a floating "0.0" below the miRNA-K12-9 label.

We appreciate the reviewers taking time to provide us with comments. Below we provide a point by point response. Appropriate changes have also been added to our manuscript.

Reviewer #1

The manuscript by Rajendren et al. provides the first global analysis of the A-to-I editome in PEL cells and KSHV transcriptome during KSHV latent and lytic replication. Using modified PEL cell lines, the authors collected latent and lytic GFP+ cells for deep RNA sequencing analysis to identify A-to-I edited sites using the SAILOR algorithm and further validated editing of specific sites by comparing the cDNA with genomic DNA through Sanger sequencing. Excitingly, the authors identified conserved and stage-specific A-to-I editing sites in the transcriptome of host BCBL1 and BC-3 cells and KSHV during latent and lytic reactivation. The authors further identified three previously unknown ADAR1-mediated conserved editing sites of the KSHV pri-miRNA-K12-4 transcript, two in the lower stem region and one within the seed region of mature miRNA-K12-4-3p, which affect (1) miRNA biogenesis, (2) target specificity, and (3) viral transmission. Using an established KSHV miRNA vector, the authors mutated the adenosines into guanosines within the lower stem region of pri-miRNA-K12-4 and showed a (1) significant decrease in mature miR-K12-4-3p and -5p not observed upon ADAR1 siRNA-mediated depletion. Using bioinformatics tools and luciferase assays, the authors showed that (2) editing of the seed sequence of miRNA-K12-4-3p expanded target recognition. Using the iSLK-BAC16 KSHV infection cell model with rescued edited miR-K12-4, the authors showed that (3) editing at the seed region increases infectious virion production. These findings shed light into the functional and biological significance of A-to-I editing in KSHV latent and lytic replication in PEL cells, opening the door for future applications. However, main concerns are the lack of ADAR1 isoform detection that may impact data interpretation, the importance of seed region editing of mature miRNA-K12-4 for KSHV transmission, lack of in-depth discussion of host and KSHV transcriptome remodeling during viral stages, and lack of percent editing in the Sanger trace to showcase the population of RNA edited.

Additional experimentation is required to support the reported findings.

1. **Comment:** *The authors do not take into consideration the reported isoforms of ADAR1. Based on reported literature, it is likely that in latent infection, p110 is the main isoform which is predominantly found in the nucleus and correlates with Fig 2h ADAR1 nuclear localization. Upon lytic reactivation, the IFN-inducible p150 isoform may be expressed which localizes into the cytosol and has been reported to edit more substrates and more sites than p110. The increased editing activity the authors observed may not only be due to increased gene expression but also induction of p150 isoform. The authors need to address both isoforms and show their expression in respective western blots. In lines 366-370, the authors discussed the slight localization of ADAR1 (i.e., p110) known to shuttle between the nucleus and cytosol upon stress (Fig 2h), but neglect the major cytosolic isoform, p150.*

Response: The antibody that we used can detect both the p110 and p150 isoforms. Similar to the recent data from the Damania group (PMCID: PMC7319254), we do not observe induction of the p150 ADAR1 isoform during lytic reactivation. We have now added to lines 183 – 184 indicating that we do not observe changes in the expression of the p150 ADAR1 isoform and that this is consistent with a previous report. This is also further noted on lines 389 – 391.

Furthermore, our data in Figure 2i supports that the editing increase is mediated by the p110 ADAR1 isoform as the p32 CLIP signal migrates between 100kDa and 130kDa, and no p150 p32 signal is observed.

- 2. Comment:** *In Fig 5d, the authors show that editing within the miRNA-K12-4-3p seed region impacts viral transmission. However, whether this editing is required by KSHV for transmission still needs to be further addressed. The authors previously describe how the pri-miRNA K12-4-3p is edited in the lower stem loop and this significantly decreases mature miRNA K12-4-3p expression in an anti-viral fashion. Following miRNA processing, primary miRNA editing should occur before the editing of the mature miRNA seed region, which appears to be pro-viral. The authors could include wt and edited pri-miRNA-K12-4 in their experiments to assess whether the editing of the cleaved miRNA-K12-4 is truly required for viral transmission and the findings are not artificial. On this note, the authors should also address which type of editing is preferred within the primary vs mature miRNA transcript in either latent or lytic stage of replication. Since ADAR1 p110 is in the nucleus, it is likely the predominant editor of pri-miRNA-K12-4, while p150 may be the predominant editor of miRNA-K12-4.*

Response: We thank the reviewer for this comment, and we have included Fig S2 in response. To investigate the cellular compartment of KSHV miRNA editing we performed Sanger sequencing assays on nuclear/cytoplasmic fractions from latent BCBL1 and BC3 cells (Fig S2). We can detect editing at all three sites in both nuclear and cytoplasmic fractions, suggesting KSHV miRNA editing occurs within nucleus. Moreover, quantification of editing from the Sanger results indicates there is no additional editing of the KSHV miRNA occurring in the cytoplasm and our quantification from latent and lytic infected cells suggests no significant changes in the editing levels in lytic reactivation.

The experiment suggesting to include wt and edited pri-miRNA is not feasible for multiple reasons, including that transfection of a full-length pri-miRNA is likely trip to cell intrinsic double stranded (ds)RNA sensors and activate an interferon response confounding any results. Additionally, transfection of a wt pri-miRNA would still be edited by endogenous ADAR. Our data in Fig 5 clearly show that loss of miR-K12-4 results in the production of virus that is severally attenuated in terms of its ability to establish a latent infection, and that only an edited miRNA is capable of rescuing.

- 3. Comment:** *In order to discreetly analyze the impact of A-to-I RNA editing in KSHV reactivation (latent vs lytic cycle), it would be important to include the comparison of global A-I editing sites, efficiency, and its transcriptional implication between cells that were induced for lytic cycle but were GFP- cells (bystander cells) within the same population. This analysis might provide insights of the mechanism of “spontaneous” lytic reactivation.*

Response: Thank you for this interesting comment. This study is focused on defining how latent (resting) and lytic editomes are remodeled and the biological impact of that editing on infection. How the GFP- (bystander cells) respond to lytic infection is a fascinating question, not only in terms of the editome but general biology. At present, those studies are out of the scope of this manuscript. However, part of our longer-term efforts towards investigating A-to-I editing plan to investigate “bystander cell” responses.

4. **Comment:** *The percentage of editing (peak height of G/(G+A)*100) in sanger sequencing could give a more accurate representation of the population of mRNA edited. This is particularly necessary for the miRNA editing observed (Fig 5d), where there seems to be lower editing compared to other mRNA regions (Fig 1h), which the authors need to address.*

Response: Thank you for this comment. We believe the reviewer is refereeing to Fig 4b and 4c instead of 5d, as there is no Sanger sequencing in 5d. The stoichiometry of RNA modifications is an important topic in the RNA modification field and can vary from very low to essentially 100%. In response to this comment, we now report the percentage of editing for all our Sanger sequencing analysis in Supplementary table S5 and Fig. S2c. In addition, SAILOR quantifies editing at each site transcriptome-wide and the values for every site identified are reported in Supplementary Table S1 (host) and Supplementary Table S3 (viral).

5. **Comment:** *The authors should focus on the novelty of their global analysis finding, further discussing specific host and KSHV transcript remodeling between latency vs lytic reactivation in addition to miRNA K12-3-4p editing sites and less emphasis on increased numeric values upon lytic reactivation. Gandy et al had previously reported that ADAR1 A-to-I editing controls the function of the Kaposin A/mi-K10 portion of the K12 transcript in PEL cells and that editing increased 10-fold by activation of lytic replication.² On this note, Zhang et al. also showed that ADAR1 facilitates KSHV lytic reactivation by modulating the RLR-dependent signaling pathway.³ The field also knows that ADAR1 preferentially edits intronic and untranslated regions mainly in host Alu elements.^{1,4}*

Response: We thank the reviewer for this comment. As this is the first comprehensive analysis of host and viral editing in PEL and the KSHV lifecycle we believe it is important to include the global quantitative view of editing. Particularly, as the data here can serve as a great resource for future investigations into the biology of A-to-I editing during KSHV infection.

Thank you for pointing out these references. In fact, in our previous manuscript all of these were cited. However, we now include additional comments for each citation highlighting points made by the reviewer (lines 236, 441-442, and 443-445).

6. **Comment:** *Lines 371-385, the authors discuss the potential of editing in protein coding regions. However, this study showcases how most host editing occurred in intronic (>60%) and untranslated regions (~30%) and focus on non-coding KSHV miRNA. The authors showed Kaposin and ORF50 (Fig 3c) editing in BCBL-1 cells, but this requires further follow up studies to prove adaptive RNA editing. On this note, unlike host analyses (Fig 1d), the authors did not detail the percent editing in different types of KSHV RNA, which could strengthen their interpretations and add to their findings.*

Response: Our studies provide a deep and comprehensive analysis of the A-to-I RNA editing landscape in PEL and the effect of editing on KSHV miRNA biology. The potential for RNA editing to influence the proteome is well established, both in humans and in other organisms, and we agree that if investigators seek to test the effect of protein recoding on the KSHV lifecycle additional studies will be required. Indeed, while proteome recoding is clearly out of the scope of this manuscript, it is a topic that we are actively investigating.

Regarding annotation of the host vs viral genome, the KSHV genome is not as well annotated. For example, KSHV transcriptome directed studies continue to identify additional introns and promoter regions. Therefore, mapping to well annotated features, such as the ORFs, miRNA, and repeat regions, is the clearest approach and best for the community currently.

Minor comments:

1. **Comment:** *Line 54, as per the Wiley citation, it is small nucleolar RNA (snoRNA) not snRNA and the authors mention they promote 2'O-methylation of adenosine which impairs editing. As per the Nishikura citation, the authors report that adar1p110 accumulates in the nucleolus proposed to binding to rRNA or to small nucleolar RNA.*

Response: This is now corrected now, and we included long non-coding RNAs and microRNAs as examples of reported ADAR substrates with appropriate references.

2. **Comment:** *Line 59, ADAR1 KO mice not only have reduced adar1 editing, but the phenotype is also lethal, mainly through MDA5-mediated mechanism*

Response: We agree. We have now added an additional sentence that ADAR KO mice are lethal due to MDA5 and RNaseL mediated mechanisms (lines 63-64).

3. **Comment:** *Line 358, it has been reported greater Z-RNA repertoire allow p150 rather than p110 to bind to more RNA and have increased activity.*

Response: We thank the reviewer for this comment. We do not observe see induction of ADAR1 p150 isoform during reactivation and our ADAR1 CLIP gel suggests that ADAR1 p110 isoform binds more RNA (Fig 2i).

4. **Comment:** *Line 380, please further describe and cite what is meant by adaptive RNA editing.*

Response: We thank the reviewer for this comment. We have now expanded our discussion to include additional information, and clarify adaptive editing as RNA editing that occurs due to a stimuli and provides an advantage to the organism/microbe. Additional references are also provided (lines 416-418).

5. **Comment:** *Line 408, authors mention five PEL lines tested for conserved editing sites in pri-miR-K12-4 yet only four PEL lines are mentioned in line 244: BC-1, BC-3, BC-5, and JSC-1.*

Response: Thank you for the comment. Typo has been corrected.

6. **Comment:** *F1 b, may be unnecessary. F1 d, donut charts are simple to follow but the order and arrows into protein coding and non-coding regions needs to be simpler to follow. They also have a subset between latent and lytic which makes it more complicated. F1 e-f, can be supplemental.*

Response: We thank the reviewer for these comments. We prefer to keep Fig 1b as it highlights the distinction between adenosine and inosine to the reader. Fig 1d, we have tried to plot different ways, and this is the most suitable way to plot our high throughput identification. However, we have modified the figure to clarify coding vs

noncoding editing. Fig 1 e-f, we think it is important to show that our host editing sites have consistent neighbor preferences and preferentially found within Alu repeats reported in previous studies.

7. **Comment:** *Fig 1h, there needs to be an average percent editing based on peak height*

Response: Thank you for this comment. We have reported the percentage of editing for all our Sanger sequencing analysis in Supplementary Table S5 and Fig. S2c. In addition, SAILOR based quantification for all sites identified are in Supplementary Table S1 and S3.

8. **Comment:** *Fig 1b, the text (lines 103-104) describes how we can compare cDNA sequence and reference genomic sequence to identify A-to-I RNA editing sites. However, figure 1B shows how adenosine is converted to inosine at the RNA level. The text and figure do not match well.*

Response: The text has been corrected to match the Figure.

9. **Comment:** *Fig 1d, may need to check the % of distribution of non-coding transcription again. The number of genes expressed in BCBL1 and BC-3 were different, but the % of distribution of the latent and lytic in the two cells showed the same percentages.*

Response: Thank you for catching this. We have corrected the Typo.

10. **Comment:** *Fig3e, y axis is ADAR activity, not ADA.*

Response: Thank you for the comment. We are quantifying an enzymatic activity (i.e. adenosine deamination) with is not specific to an ADAR. The y axis is now labeled as Adenosine Deaminase Activity.

11. **Comment:** *Fig 3c, were BC-3 cells used to see the editing sites within Kaposin A?*

Response: We have added BC3 Sanger sequencing data to Fig 3c.

12. **Comment:** *Fig 4e, what should be the size of these bands? The size of the KSHV miRNA vector with G seems to be lower than the KSHV miRNA vector with A.*

Response: Thank you for the comment. The sizes of the bands have now been added to the Northern blot. The miRNAs migrate at the same size, however, the intensity of the miRNA produced from the A vector is stronger, which is quantified on the right.

13. **Comment:** *Fig 4b, Two "BC-3" texts overlap each other.*

Response: Thank you for the comment. This has been corrected.

14. **Comment:** *Fig 4e, Extra box presents in the key section.*

Response: Thank you for the comment. It has been corrected.

15. **Comment:** *Fig 2f, MW are missing on the western blot*

Response: We have now added the MW.

16. **Comment:** *Fig 4d and Fig 5c, need improved replicated for less variation and accurate statistics.*

Response: We have now repeated the experiment with an additional biological triplicate setting. The new data is included in the revised figures. Our previous conclusion stands.

17. **Comment:** *Fig S1c, need labels for BCBL1 and BC-3 cell type.*

Response: Thank you for the comment. Labels have been added.

Reviewer #2:

In this paper, the authors explore the roles of A-to-I editing in KSHV-infected cells. They first catalog A-to-I editing within latent and lytic cells and show that the A-to-I editome changes between phases. They examine both host and viral gene targets and then delve more deeply into the effects of KSHV miRNA editing on processing and function. Overall, the data are clearly presented and interpreted appropriately. The findings impact both the KSHV and RNA editing fields, so it has a broad audience. I have only minor comments for improvement of the manuscript:

1. **Comment:** *Line 97: "We collected latent and lytic GFP + cells". This reads as if latent cells are GFP positive. Please clarify this sentence in the text.*

Response: We thank the reviewer for this comment. The sentence has been corrected.

2. **Comment:** *Please label the two graphs in Figure S1c on the diagrams (BCBL1 and BC3). Similarly, label Fig 2l/m on graphs.*

Response: Thank you for catching this. Labels have been added.

3. **Comment:** *X-axes are often labeled log2fold. Log2-fold of what? Some quantity should be given in the label.*

Response: Thank you for the comment. Log2fold of normalized read counts from our RNA-seq data was presented in our figures. Labels have been corrected.

4. **Comment:** *Color code key should be given for the Sanger sequencing traces.*

Response: Thank you for suggesting this. A color code key was added to the figure legends.

5. **Comment:** *In Fig 4a, please graphically depict the sequence of the mature miRNAs and seed sequences on the pri-miRNA. It would be helpful for readers to know where these features are to compare them to the sites of editing.*

Response: We thank the reviewer for this comment. Mature miRNA sequences are graphically depicted in our revised pri-miRNA-K12-4 schematic with seed sequences of both 3p and 5p miRNAs marked in addition to the edited sites.

6. **Comment:** *In Fig 4d the miRNA is labeled K4 instead of K12-4.*

Response: Thank you for the comment. Typo has been corrected.

7. **Comment:** *For the data presented in Fig 4k, they state that they “cloned the 3UTR of six predicted targets...”. How are they defining “predicted” here? Does this mean predicted to be targeted by both, edited, or unedited miRNAs? If they are specific, do the results correlate with predictions?*

Response: Thank you for the comment. We have now clarified how we selected targets on lines 307 - 311. Specifically, we selected targets that are regulated by both unedited and edited miRNA (WNT3, TAB2), targets that are preferentially regulated by unedited miRNA (HECTD2 and TPD52) and targets that are preferentially regulated by edited miRNA (EGR1 and ROCK2). The results of the assay confirm our predictions and demonstrate that editing impacts target selection.

8. **Comment:** *The authors correctly state that the data in Fig 5c for the complementation with “miRNA with G” are not statistically significant. However, 2 replicates are quite a bit higher than the third and the lower mean and higher error is due to the third one replicate. This may point to biological relevance, particularly in light of the results in Fig 4d. How confident are they that the lytic replication is not better in these cells? This should perhaps be noted.*

Response: We have now repeated these experiments in triplicate and observe less variation in the new data. The new data is included in the revised Fig 5C and suggests that the number of viral genomes in the supernatant collected from the complementation experiment is not significantly different between our five samples.

9. **Comment:** *Overall, the labels on figures tend to be too small.*

Response: The size of the labels has been adjusted.

Reviewer #3:

This manuscript reports the A to I editing patterns of cells infected with KSHV virus, and in either the latent or reactivating state of infection. They demonstrate that there is a substantial increase in A-I editing of the KSHV genome and of the cellular genome in cells undergoing lytic reactivation, and that the A-I editing pattern in these 2 states of infection has some overlap, but also has very clear distinctions in the A-I edited sites. These findings are exciting and novel, and provide insights into an important source of RNA editing during infection, with potential for biological outcomes during infection. They go on to show that one of the edited sites in a viral miRNA results in a functional change in the miRNA, such that the result is a

significant decrease in virus lytic reactivation. These new insights are strong and are important to the fields of RNA biology and to virus-host interactions.

The major conclusions and large datasets reported in this manuscript (above) are sound, yet there are some secondary conclusions and details that are not strong enough to support the conclusions. Particularly, the authors have not demonstrated that modification of the miRNA-K-12-4 sequence results in downstream defects in virus binding and entry.

1. **Comment:** *Several experiments use transfected/modified cells that are not well-described and for which there is no validation data provided. In nearly every case, the pedigree and timing of the cells should be clarified for the reader, potentially by timeline schematics or tables.*
 - a) *the pPAN reporter cells are very briefly described, and are essential to the study. Data should demonstrate their inducibility relative to parent lines and their GFP status with and without induction. Evidence that the sorts result in pure populations of latent vs lytic reactivating cells could be further supported by the RNA Seq files. Flow cytometry data should include stained and unstained samples, controls, live/dead discrimination, gating strategies, and comparison of pre-sort and post-sorted cells. Clarification on whether 'latent cells' refer to untreated cells or cells sorted and selected as GFP-negative. The same goes for the PAN fish flow staining, PAN anti-sense oligos should be listed, and methods are unclear on which Alexafluor is used in experiments shown.*

Response: We thank the reviewer for this comment and we have added significant description to our pPAN reporter cells. We now include Fig S3 with the gating strategies with control cells for our Flow cytometry. Fig S3 contains a table representing our normalized RNA-seq read counts in our latent (untreated) and lytic (48 hpi) samples. Latent refers to the untreated samples and we have added this information to our manuscript in lines 105 and 512 and to the figure legends at the appropriate places. PAN-antisense oligo is listed in Supplementary Table S4.

b)Comment: *The iSLK and BCBL-TRex cell transfection schemes should be clarified. There are a number of miRNA and mimics and each could be made more plain for the reader. Text and Methods differ in transfection methods.*

Response: Thank you for this comment. We added more details to the iSLK and TRex-BCBL1 transfection schemes for clarification.

2. **Comment:** *While the manuscript does a very convincing demonstration of A-I editing distinctions and shows quite well that A-I editing of a viral miRNA compromised virus lytic reactivation, there are subsequent studies of the resulting virus that are not well supported. In Figure 5d, the authors show that miRNA-K12-4 mimics with sequences matching the edited or unedited sequences result in striking differences in virus reactivation with the result of significantly decreased virus production. However, in Figure 5f-h, the data shown is taken to mean that the resulting virus from this supernatant is itself defective in subsequent binding and entry. If this were to be proven, it would require inoculation of equivalent amounts of virus for analysis of binding and entry. This is a significant burden of proof, and could be quite difficult given the very poor virus production shown in Figure 5c/d. Methods state that these cells were infected with an MOI=3 particles, but no particle measurements*

can be made without EM or Virocyte, so it is unclear what is meant here. Viral DNA is reported as relative to WT rather than relative to input, and no limit of detection or background level is provided. As is presented, there is not convincing data shown to refute the reasonable conclusion that reduced virus production results in reduced virus available for binding and entry.

Response: We apologize for the confusion as there is some misinterpretation in the way we presented panels in Fig 5. To aid in clarification we have modified Fig 5 and corrected (i.e.) removed particles from MOI as it was a typo.

In Figure 5b and 5c we demonstrate that viral reactivation (5b, gene expression; 5c virion production/genome numbers) are not affected by loss of miR-K12-4. We now clearly indicated 5b is lytic gene expression, and 5c is now viral genomes/mL. In Figure 5d, which is the supernatant transfer assay, loss of miR-K12-4 results in a significant reduction in infectious virions, that can only be rescued by providing the edited miR-K12-4. We now modify the schematic to indicate it is the supernatant transfer. To investigate virus binding and entry we performed assay's similar to those previously described in (PMCID: PMC3894220), in which we used a similar number of viral genomes for binding and entry assays.

3. **Comment:** *The schematic refers to virus infection as transmission, however, virus transmission is the process by which viruses spread between hosts.*

Response: Thank you for this comment. We used transmission as we were transmitting the virions produced from iSLK-bac16 cells to another cell line (either HEK 293T or HUVECs). However, we have now replaced 'transmission' with 'infection' in the revised manuscript.

4. **Comment:** *Figure 2h, there is some H3 signal in the cytoplasmic fractions. Comment on whether this relates to virus infection and nuclear membrane integrity?*

Response: Fractionation of B cells is challenging given their small size and we occasionally observe a minor contamination of H3 in our cytoplasmic fractions. Since this occurs in both latent and lytic infected cells, this reflects the technical difficulty of nuclear cytoplasmic fractionation of PEL cells.

5. **Comment:** *Figure 2j/k: comment on relatively rare events beyond 2-fold changed?*

Response: We apologize for the oversight. As the majority of the expression level changes are at or below 2-fold changes, the x-axis was restricted to 2-fold changes in the previous figure. We have corrected this and extend the x-axis to 8-fold change in expression and all data points are visible on this new version.

6. **Comment:** *Figure 4e has an unlabeled white box below "KSHV miRNA vector with G", and Figure 4e has a floating "0.0" below the miRNA-K12-9 label.*

Response: Thank you for this comment. Both are corrected in the revised figures.

Reviewers' Comments:

Reviewer #1:

Remarks to the Author:

In this revision, the authors defined the ADAR1 isoform involved, ADAR1 p110, and included a report of the editing percentage of their Sanger sequencing analyses in addition to SAILOR quantification of both host and viral edited sites. The authors also addressed their longer-term efforts to investigate GFP- bystander cell responses, included the multiple references to be added, and addressed the setbacks of attempting to annotate the KSHV editing genome and incorporating edited pri-miRNA. The authors incorporated all minor comments, including figure modifications, labeling, figure description as well as additional points of discussion, references, and necessary replicates. Overall, the authors have addressed all major comments and their additional experimentation and revision improved their original submitted manuscript. I only have additional minor comments.

Minor comments

- Figure 1h and Figure 4b. The Sanger trace seem as if they were the same figure repeated for the different cell lines genomic cDNA traces for one target. If the figures are just a representation of genomic cDNA target, including only one can make interpretations clearer.
- Figure 4L. Legend for dark blue is missing.

Reviewer #3:

Remarks to the Author:

This manuscript regarding ADAR editing patterns of KSHV infected cells, in either latent (untreated) or reactivating state of infection. They demonstrate that there is a substantial increase of A-I editing of both KSHV and the cellular genomes in cells undergoing lytic reactivation, and demonstrate distinct edited sites. These findings are exciting and novel, and provide insights into an important source of RNA editing during infection.

This revised manuscript has addressed the majority of reviewer concerns, however, there remain outstanding details. These are critical to reader understanding and to the potential for others to replicate the study or to implement methods or reagents. These are essential parts of any publication and can be easily revised to meet standards for rigor and authentication.

Methods section for Cells and Viruses should contain ALL cells (293, HUVEC, BC3, TRex-BCBL1, iSLK etc) and all viruses used in the paper, along with a source or reference for each, or a description of how the cells were generated if new to this study, and what they are subsequently called.

PAN-GFP reporter cells are named differently in Cells and Viruses than in siRNA KD section of Methods and should be consistent.

BC3 cells do not contain a Dox inducible cassette- how were they induced? (This is needed in Methods, figure, legend, text.)

FISH-Flow description that is now included is better, but still missing details.

How was the FISH-Flow oligonucleotide labeled (in house or commercial, purification from unlabeled?, end-labeling, direct conjugation, which Alexaflour probe of the two mentioned was used, or if both then which in which settings)?

Line 476 should reference Supp Table 4 which lists the PAN oligo.

Line 478 should define "extensive" washing- how many and/or in what volume?

While RNA Seq data shown for purified latent and lytic cells show that the sort purification was relatively successful, use of a single 24-mer oligo for FISH-Flow has a very high probability to have high background and the 24-mer does show identity with other human sequence. While this worked adequately, there should be a mention of the caveats of this approach. For example, the BCBL1 latent cells show background GFP signal and the latent BC3 cells show background PAN signal, and what rescues the sort strategy is only selecting the double positives. It is important for the authors to point this out to the reader and to those who may adopt a method that is not sufficient as a single marker.

Figure S3 requires a full legend. What is the GFP control, PAN control, negative controls, x-axis label missing under upper left panel, what is APC-A (which PAN Alexaflour?), etc.

Addition of FigS3B provides critical support for the validity of the cells and sorts- great!

Figure 5H still refers to "transmission" which should be changed or tempered because it is not

proven. However, the idea that ADAR editing could impact transmission is interesting, and could be included in discussion on possibilities.

The revised manuscript now referenced "MOI=3", but should include units (3 what?) such as genome equivalents or what best describes virus quantitation here.

Reviewer #1

In this revision, the authors defined the ADAR1 isoform involved, ADAR1 p110, and included a report of the editing percentage of their Sanger sequencing analyses in addition to SAILOR quantification of both host and viral edited sites. The authors also addressed their longer-term efforts to investigate GFP- bystander cell responses, included the multiple references to be added, and addressed the setbacks of attempting to annotate the KSHV editing genome and incorporating edited pri-miRNA. The authors incorporated all minor comments, including figure modifications, labeling, figure description as well as additional points of discussion, references, and necessary replicates. Overall, the authors have addressed all major comments and their additional experimentation and revision improved their original submitted manuscript. I only have additional minor comments.

Comment: *Figure 1h and Figure 4b. The Sanger trace seem as if they were the same figure repeated for the different cell lines genomic cDNA traces for one target. If the figures are just a representation of genomic cDNA target, including only one can make interpretations clearer.*

Response: Thank you for this comment and we really appreciate it. After careful consideration, we still want to report the genomic DNA traces from every cell line we assessed the editing sites to show the reader that the genomic traces are purely adenosines at the edited sites. This will eliminate any concern the reader might have on whether the genomic DNA from different cell lines contains SNPs.

Comment: *Figure 4L. Legend for dark blue is missing.*

Response: We thank the reviewer for this comment and the legend was added.

Reviewer #3

This manuscript regarding ADAR editing patterns of KSHV infected cells, in either latent (untreated) or reactivating state of infection. They demonstrate that there is a substantial increase of A-I editing of both KSHV and the cellular genomes in cells undergoing lytic reactivation, and demonstrate distinct edited sites. These findings are exciting and novel and provide insights into an important source of RNA editing during infection.

This revised manuscript has addressed the majority of reviewer concerns, however, there remain outstanding details. These are critical to reader understanding and to the potential for others to replicate the study or to implement methods or reagents. These are essential parts of any publication and can be easily revised to meet standards for rigor and authentication.

Comment: *Methods section for Cells and Viruses should contain ALL cells (293, HUVEC, BC3, TRex-BCBL1, iSLK etc) and all viruses used in the paper, along with a source or reference for each, or a description of how the cells were generated if new to this study, and what they are subsequently called.*

Response: We thank the reviewer for this comment. We have included all the cell lines used in this study and their growth conditions in our methods section in lines 471-523. We also expanded where we describe how we generated the two new PEL cell lines that are used in this study in the method section to make it clear to the reader (lines 481-521).

Comment: *PAN-GFP reporter cells are named differently in Cells and Viruses than in siRNA KD section of Methods and should be consistent.*

Response: Thank you for this comment. We have corrected the naming in the siRNA KD section to maintain consistency.

Comment: *BC3 cells do not contain a Dox inducible cassette- how were they induced? (This is needed in Methods, figure, legend, text.)*

Response: Thank you for this comment. Related to this reviewer's first comment, this is now more clearly stated in the methods where we describe the generation of Dox-inducible BC3-PAN-GFP cells (lines 481-521). We also report this in the text on lines 99-102, and this has been included in the figure legend.

Comment: *FISH-Flow description that is now included is better, but still missing details. How was the FISH-Flow oligonucleotide labeled (in house or commercial, purification from unlabeled?, end-labeling, direct conjugation, which Alexaflour probe of the two mentioned was used, or if both then which in which settings)?*

Response: We have added the additional information to our methods section and describe how the FISH-Flow oligonucleotide was commercially acquired as well as the details of fluorophore location.

Comment: *Line 476 should reference Supp Table 4 which lists the PAN oligo.*

Response: Thank you for this comment. We added Supplementary Table S4 as a reference for PAN FISH-flow oligonucleotide sequence.

Comment: *Line 478 should define "extensive" washing- how many and/or in what volume?*

Response: We thank the reviewer for this comment, and indicate that sample was washed twice.

Comment: *While RNA Seq data shown for purified latent and lytic cells show that the sort purification was relatively successful, use of a single 24-mer oligo for FISH-Flow has a very high probability to have high background and the 24-mer does show identity with other human sequence. While this worked adequately, there should be a mention of the caveats of this approach. For example, the BCBL1 latent cells show background GFP signal and the latent BC3 cells show background PAN signal, and what rescues the sort strategy is only selecting the double positives. It is important for the authors to point this out to the reader and to those who may adopt a method that is not sufficient as a single marker.*

Response: We thank the reviewer for this comment and we really appreciate it. We now have added more information about caveats of using one selection and the advantage of selecting double positives (lines 107 – 108).

Comment: *Figure S3 requires a full legend. What is the GFP control, PAN control, negative controls, x-axis label missing under upper left panel, what is APC-A (which PAN Alexaflour?), etc.*

Response: Thank you for this comment and we have added a full legend to Figure S3.

Comment: *Figure 5H still refers to "transmission" which should be changed or tempered because it is not proven. However, the idea that ADAR editing could impact transmission is interesting, and could be included in discussion on possibilities.*

Response: We thank the reviewer for this comment. It has been corrected from "transmission" to "infection" in Figure 5H to match the text.

Comment: *The revised manuscript now referenced "MOI=3", but should include units (3 what?) such as genome equivalents or what best describes virus quantitation here.*

Response: We thank the reviewer for this comment and we have added "genome equivalents" as the unit.